# A diverse intrinsic antibiotic resistome from a cave bacterium

Andrew C. Pawlowski[1], Wenliang Wang[1], Kalinka Koteva[1], Hazel A. Barton[2], Andrew G. McArthur[1] & Gerard D. Wright[1]

Antibiotic resistance is ancient and widespread in environmental bacteria. These are therefore reservoirs of resistance elements and reflective of the natural history of antibiotics and resistance. In a previous study, we discovered that multi-drug resistance is common in bacteria isolated from Lechuguilla Cave, an underground ecosystem that has been isolated from the surface for over 4 Myr. Here we use whole-genome sequencing, functional genomics and biochemical assays to reveal the intrinsic resistome of *Paenibacillus* sp. LC231, a cave bacterial isolate that is resistant to most clinically used antibiotics. We systematically link resistance phenotype to genotype and in doing so, identify 18 chromosomal resistance elements, including five determinants without characterized homologues and three mechanisms not previously shown to be involved in antibiotic resistance. A resistome comparison across related surface *Paenibacillus* affirms the conservation of resistance over millions of years and establishes the longevity of these genes in this genus.

[1] Michael G. DeGroote Institute for Infectious Disease Research and the Department of Biochemistry and Biomedical Sciences, McMaster University, Hamilton, Ontario, Canada L8S 4K1. [2] Department of Biology, University of Akron, Akron, Ohio 44325, USA. Correspondence and requests for materials should be addressed to G.D.W. (email: wrightge@mcmaster.ca).

Understanding the evolution and origins of antibiotic resistance genes is vital to predicting, preventing and managing this global health problem. Studies over the past decade have revealed that antibiotic resistance is common in contemporary and ancient environmental bacteria, and that the associated genes are similar or identical to those circulating in pathogens[1]. Isolation of metagenomes from samples of 5,000–30,000-year-old permafrost reveals an abundance of resistance elements able to protect against penicillin, tetracycline, aminoglycoside and glycopeptide antibiotics[2,3]. This is consistent with surveys of modern non-pathogenic environmental bacteria and metagenomes that show that the antibiotic resistome, the collection of all antibiotic resistance elements, is extensive in the microbial world, as well as genetically and mechanistically varied[4–8]. Antibiotic resistance and the resistome are therefore ancient, highly diverse and globally distributed.

While resistance is a natural phenomenon, what has changed since the beginning of the antibiotic era is a marked increase in selection for mobile resistance elements and consequently, accretion of drug resistant pathogenic and non-pathogenic bacteria. Furthermore, the diversity of resistance in individual strains and the combination of multiple genes on a single genetic platform are reaching new heights. Some bacteria harbour dozens of acquired resistance elements often conferring redundant protection against individual antibiotics. It is clear that where antibiotic use is abundant, a plethora of resistance elements is expected. This is consistent with the extensive intrinsic resistomes of soil bacteria, such as actinomycetes that also produce many antibiotics[4,5]. Soil bacteria are rich in resistance elements because there are many antibiotic producers in the soil; similarly, acquired antibiotic resistance is common in bacterial strains causing disease in intensive care units, manured soils, and other clinical and agricultural sites where antibiotic use is intense[9–12].

In a previous survey of antibiotic resistance in a panel of bacterial strains isolated from Lechuguilla Cave, New Mexico, we found that multi-drug resistance was common in Gram-negative and Gram-positive bacteria. The deeper portions of Lechuguilla Cave have been isolated from the surface for over 4 Myr, with no animals or metazoans living in this environment[13,14]. The energy that powers the cave microbial ecosystem is largely acquired through chemolithrotrophic means, with no exposure to exogenous antimicrobial compounds. In our initial study of Lechuguilla Cave bacteria, we identified *Paenibacillus* sp. LC231 as a strain showing a remarkable drug resistance phenotype that included the lipopeptide daptomycin, the last novel antibiotic scaffold to be introduced to the clinic. Here we systematically probe the intrinsic resistome of *Paenibacillus* sp. LC231 through whole-genome sequencing and functional genomics, and identify several new resistance elements. The result is a comprehensive analysis of resistance genotype and phenotype from a bacterial strain isolated geologically and temporally from contemporary surface members of the same genus and species. We report both a remarkable conservation of resistance mechanisms over millions of years and also the discovery of five new resistance elements that we now realize are widespread in the environment and thus potential sources of clinical concern should they be mobilized into pathogens.

## Results

**Paenibacillus sp. LC231 is multi-drug resistant**. We generated a quantitative antibiogram for *Paenibacillus* sp. LC231 for 40 different antibiotics that target diverse cellular processes and compared it with the susceptibility of a surface strain of *Paenibacillus lautus* ATCC 43898, the pathogen *Staphylococcus aureus* RN4220 and the environmental bacterium *Kocuria*

*rhizophila* (formerly identified as *Micrococcus luteus*) (Supplementary Tables 1 and 2). *Paenibacillus* does not have specified minimal inhibition concentration (MIC) values to define resistance; therefore, in this study, resistance is relative to *K. rhizophila* and *S. aureus*. Further experiments in this study investigate the molecular basis of resistance and not the contribution of each genetic determinant to the resistance phenotype of *Paenibacillus* sp. LC231. Both species of *Paenibacillus* had MICs at least 8× higher than *K. rhizophila* or *S. aureus* for 26 antibiotics and, in general, were significantly more resistant to these antibiotics. We focused our investigation on *Paenibacillus* sp. LC231 and determined that in addition to resistance to 14 classes of antibiotic, it inactivates 7 distinct classes.

Sequencing the *Paenibacillus* sp. LC231 genome followed by analysis of resistance genotype using the Comprehensive Antibiotic Resistance Database (CARD)[15] enabled comparison of the resistance genotype with experimentally determined phenotype (Fig. 1; Supplementary Table 1). These correlated well for many antibiotic classes. For example, decreased rifampin sensitivity was consistent with mutation in the drug target RpoB (Ala473Thr) and the presence of a predicted rifampin phosphotransferase-encoding gene (*rph*) that may inactivate these drugs by

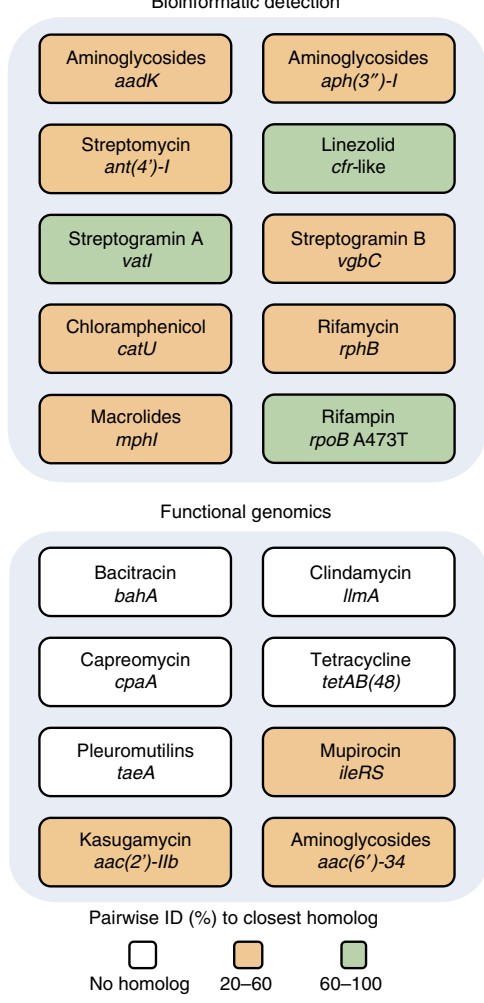

**Figure 1 | The antibiotic resistance genotype of *Paenibacillus* sp. LC231.** The Comprehensive Antibiotic Resistance Database (CARD) annotated nine resistance genes and one resistant variant from the draft genome sequence. For phenotypes where a genotype could not be identified, functional genomics was employed, identifying eight resistance elements, including five novel gene families.

phosphorylation of the hydroxyl at C21 of the *ansa* chain[16]. Our informatic analysis identified an orthologue of the 23S ribosomal RNA (rRNA) methyltransferase Cfr (95% identical to the characterized Cfr-like enzyme ClPa from *Paenibacillus* sp. Y412MC40 (ref. 17)), which is consistent with slightly decreased sensitivity to linezolid, pleuromutilin, streptogramin A, lincosamide and chloramphenicol antibiotics. In addition to the Rph, our analysis further predicted that the *Paenibacillus* sp. LC231 genome encodes several orthologues of antibiotic-inactivating enzymes: a chloramphenicol acetyltransferase (Cat), a streptogramin B lyase (Vgb), streptogramin A acetyltransferase (Vat) and a macrolide kinase (Mph), all of which confer drug resistance when heterologously expressed in *Escherichia coli* (Table 1; Supplementary Table 3). We overexpressed and purified each enzyme and confirmed their catalytic activity using high-performance liquid chromatography (HPLC) and mass spectrometry (MS; Supplementary Fig. 1; Supplementary Table 4).

We were unable to match some of the antibiotic resistance phenotypes of *Paenibacillus* sp. LC231 to specific genes using CARD, including resistance to bacitracin, clindamycin, capreomycin, mupirocin, tetracycline, tiamulin, kanamycin and kasugamycin. These resistance determinants could be the results of novel genes or significantly divergent from known mechanisms such that CARD did not recognize them. We therefore constructed a *Paenibacillus* sp. LC231 genomic library in *E. coli* and selected for clones that confer resistance to these antibiotics in this heterologous host (Fig. 1; Table 1; Supplementary Tables 1, 3 and 4). Using this functional genomics strategy, we identified a putative isoleucyl-transfer RNA synthetase conferring resistance to mupirocin, a predicted 23S rRNA methyltransferase that confers resistance to clindamycin, and enzymes that inactivate capreomycin, kanamycin and kasugamycin, respectively. We identified the bacitracin resistance gene by screening clones for bacitracin-modifying activity using an overlay technique. *E. coli* is naturally insensitive to bacitracin and therefore we overlayed bacitracin-sensitive *K. rhizophila* over the *E. coli* housing the genomic library to identify bacitracin-inactivating activity. These new resistance elements are discussed below. We also identified two predicted ABC-transporters that confer resistance to tetracycline (TetAB(48)) and tiamulin (TaeA). All resistance determinants were found on the *Paenibacillus* sp. LC231 chromosome and there is no evidence of nearby mobile elements.

**BahA is an amidohydrolase that inactivates bacitracin.** Bacitracin inhibits cell wall biosynthesis by sequestering the lipid II carrier, undecaprenyl pyrophosphate and prevents it from being recycled. The antibiotic forms six hydrogen bonds with the pyrophosphate of undecaprenyl pyrophosphate; five that are made by the peptide backbone and one that is made by the amido side chain of asparagine-12 (Fig. 2a,b). High-resolution MS established that the inactivated product had a mass increase of 0.9844 Da over bacitracin (Fig. 2c,d). This change in mass is consistent with hydrolysis of –NH$_2$. We determined the structure of the inactivated product using tandem MS and revealed that asparagine-12 was hydrolysed to an aspartate (Fig. 2c; Supplementary Table 5).

The functional genomic strategy designed to identify antibiotic-inactivating enzymes led to the three unique bacitracin-inactivating clones. Sequencing each of the fragments revealed a single common gene encoding a hypothetical membrane protein (BahA) that is related to the PF12695 family (alpha/beta-hydrolase, e-value: $7.1 \times 10^{-14}$). We demonstrated that *bahA* confers bacitracin resistance when heterologously expressed in *E. coli* BL21(DE3) (Table 1) and inactivates bacitracin consistent with amidohydrolysis (Supplementary Table 4). BahA is predicted to be made up of two domains; an N-terminal domain with five transmembrane helices and a large C-terminal hydrolase domain.

**CpaA is a novel capreomycin acetyltransferase.** Functional genomic assays to identify capreomycin resistance elements identified one unique genome fragment, encoding a hypothetical protein that is related to GCN5 family of acetyltransferases (PF13523; e-value: GNAT_1 acetyltransferase, $8.18 \times 10^{-6}$) without a known homologue. In an enzyme assay using purified CpaA the mass of inactivated capreomycin product was consistent with acetylation, confirming that this enzyme is a capreomycin acetyltransferase (Supplementary Table 4). Capreomycin inhibits protein synthesis by binding to the interface of the 50S and 30S ribosomal subunits, preventing translocation. Capreomycin acetylation has previously been shown to be catalysed by the unrelated mycobacterial Eis enzymes; promiscuous enzymes that acetylate several, structurally distinct substrates with free amines, including capreomycin, aminoglycosides and ciprofloxacin[18]. Acetylating the ε-amino group of the

**Table 1 | MICs of antibiotic resistance determinants heterologously expressed in *E. coli*.**

| Resistance Determinant | Name | Antibiotic class | Antibiotic | MIC (µg ml$^{-1}$) | Empty vector control MIC (µg ml$^{-1}$) |
|---|---|---|---|---|---|
| Kasugamycin acetyltransferase | *aac(2′)-IIb* | Aminoglycoside | Kasugamycin | >2,048 | 256 |
| Kanamycin acetyltransferase | *aac(6′)-34* | Aminoglycoside | Kanamycin | >128 | 8 |
| Capreomycin acetyltransferase | *cpaA* | Tuberactinomycin | Capreomycin | >2,048 | 128 |
| Streptogramin A acetyltransferase | *vatI* | Streptogramin A | Flopristin | 256 | 16 |
| Streptogramin lyase | *vgbC* | Streptogramin B | Linopristin | | NA* |
| Chloramphenicol acetyltransferase | *catU* | Phenicol | Chloramphenicol | 256 | 4 |
| Macrolide phosphotransferase | *mphI* | Macrolide | Erythromycin | 32 | 32 |
| | | | Telithromycin | 512 | 4 |
| | | | Tylosin | 512 | 128 |
| Rifampin phosphotransferase | *rphB* | Rifamycin | Rifampin | >128 | 4 |
| Isoleucine tRNA synthetase | *ileRS* | Mupirocin | Mupirocin | >100 | 25 |
| Bacitracin amidohydrolase | *bahA* | Bacitracin | Zinc bacitracin | 512 | 256 |
| 23 S rRNA methyltransferase | *llmA* | Lincosamides | Clindamycin | 1,024 | 32–64 |
| Tetracycline ABC-transporter | *tetAB(48)* | Tetracyclines | Tetracycline | 32 | 4 |
| Tiamulin ABC-transporter | *taeA* | Pleuromutilin | Tiamulin | 1,024 | 64 |

MIC, minimal inhibition concentration, rRNA, ribosomal RNA, tRNA, transfer RNA.
*The *E. coli* MIC for linopristin is >128 µg ml$^{-1}$, the highest soluble concentration.

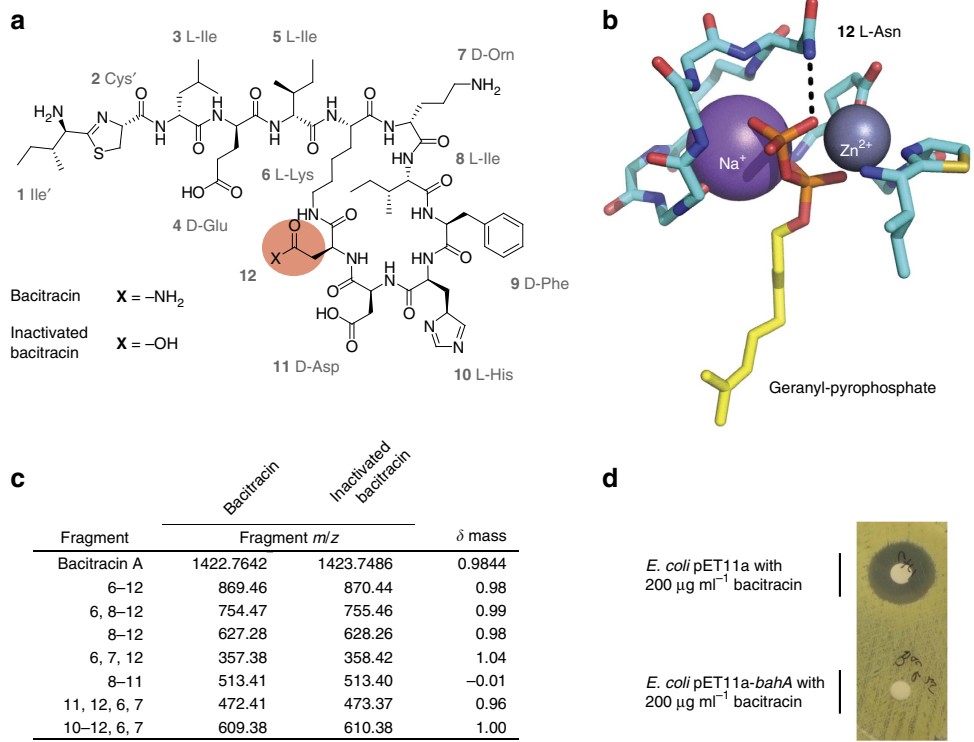

**Figure 2 | *Paenibacillus* sp. LC231 inactivates bacitracin through amidohydrolysis.** (**a**) Structure of bacitracin highlighting asparagine-12. (**b**) Structure of bacitracin A bound to geranyl-pyrophosphate (PDB 4K7T). Hydrogen bonding between bacitracin and the pyrophosphate moiety is indicated with dashed lines. (**c**) Molecular fragments of bacitracin and inactive bacitracin (R12->D12) identified using tandem mass spectrometry. The observed fragments correspond to the ring portion of the structure. Residues in each fragment correspond to the numbering in **a**. (**d**) Bacitracin inactivation by *E. coli* BL21(DE3) pET11a-*bahA* assayed using a Kirby–Bauer assay.

β-lysine by Eis confers resistance to capreomycin. To determine if CpaA specifically inactivates capreomycin, we probed for acetylation of kanamycin (an aminoglycoside) and viomycin, a structurally similar antibiotic with β-lysine on diaminopropionate at position 1 instead of 3. CpaA was unable to acetylate kanamycin or viomycin (Fig. 3a). We elucidated the structure of acetyl-capreomycin and determined that CpaA inactivates capreomycin by acetylating the α-amino group of diaminopropionic acid at position 1, which is absent in viomycin (Fig. 3b, nuclear magnetic resonance (NMR) data in Supplementary Fig. 2 and Supplementary Tables 7 and 8). Using steady-state kinetics, we determined the $K_m$ for capreomycin to be sub-micromolar ($0.76 \pm 0.12 \, \mu M$; Supplementary Table 6) with a specificity constant ($k_{cat}/K_m$) of $6.7 \times 10^5 \, M^{-1} s^{-1}$; values almost three orders of magnitude lower than those of Eis ($654 \pm 45 \, \mu M$ and $1.9 \times 10^3 \, M^{-1} s^{-1}$).

**MphI is a macrolide kinase with unique substrate specificity.** Macrolides are composed of a C14, C15 or C16 macrolactone ring and are tailored with a variety of sugars—at least one amino sugar is always on the C5 position. These antibiotics bind the ribosome at the peptidyltransferase centre and modulate translation[19]. Inactivation of this family can occur by addition of a phosphate onto the 2′ OH of the amino sugar[13,20]. *Paenibacillus* sp. LC231 is resistant to many macrolide antibiotics and we identified a potential macrolide kinase-encoding gene (*mphI*) in the draft genome. Overexpression and purification of MphI confirmed the predicted enzyme activity (Fig. 4; Supplementary Fig. 2; Supplementary Table 6). Consistent with the *Paenibacillus* sp. LC231 macrolide resistance phenotype; spiramycin, tylosin and telithromycin are inactivated, whereas azithromycin,

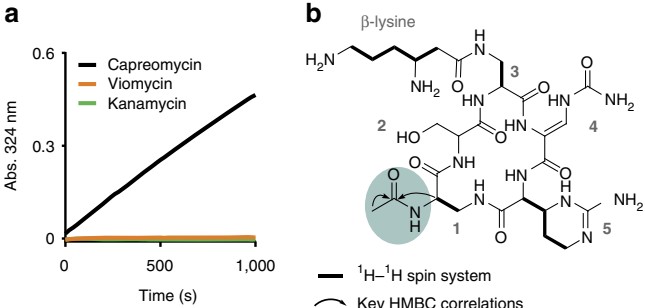

**Figure 3 | CpaA is a novel capreomycin acetyltransferase.** (**a**) CpaA specifically modifies capreomycin. A spectrophotometric assay was used to monitor acetylation of capreomycin, viomycin or kanamycin by CpaA. (**b**) Structure of capreomycin highlighting the site of CpaA acetylation. Arrows and bold bonds represent important correlations in multi-dimensional NMR experiments. Arrows represent important multi-bond heteronuclear multiple-bond correlation spectroscopy (HMBC) correlations and bold bonds represent multi-bond total correlation spectroscopy (TOCSY) and heteronuclear single-quantum correlation spectroscopy (HSQC) correlations.

clarithromycin and azithromycin are not. We hypothesized that these macrolides are not MphI substrates because they all share the presence of a C3-cladinose. We removed the C3-cladinose of clarithromycin using acid hydrolysis to produce descladinose clarithromycin (Fig. 4b; Supplementary Figs 3 and 4; Supplementary Table 9) and determined that this derivative was indeed a substrate for MphI thus confirming the narrow specificity of the enzyme (Fig. 4b–d).

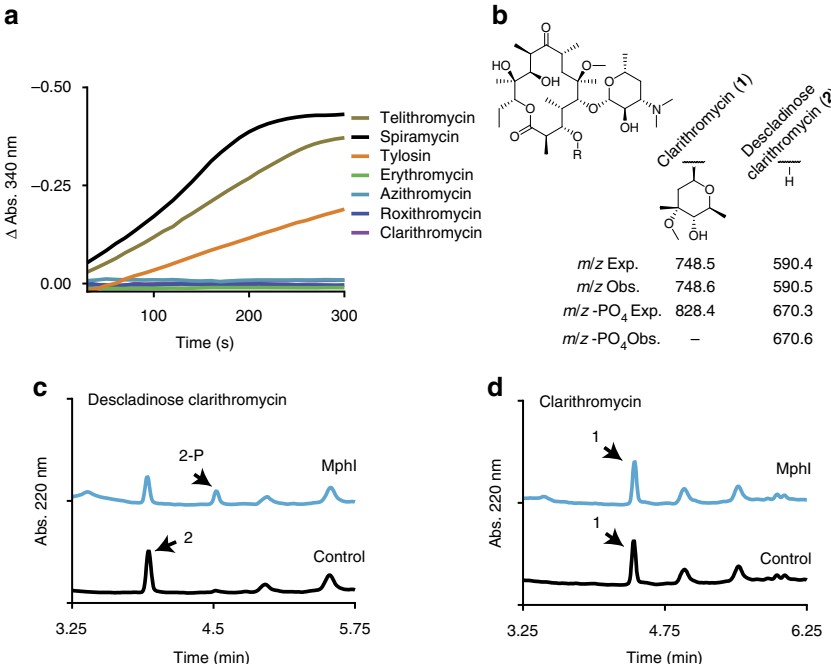

**Figure 4 | MphI is a macrolide kinase with novel substrate specificity.** (**a**) A selection of macrolides were examined for phosphorylation by MphI. (**b**) The chemical structure of clarithromycin highlighting the site of Mph phosphorylation and the C3 position. Below, the expected and observed masses for clarithromycin, descladinose clarithromycin and the phosphorylated products. (**c**) MphI phosphorylation of descladinose clarithromycin. (**d**) As a control, a similar reaction was performed using clarithromycin.

**LlmA modulates sensitivity to peptidyltransferase antibiotics.** We identified a clindamycin resistance gene that is related to the RlmK 23S ribosomal RNA methyltransferase COG family (COG1,092, $e$-value: $1.52 \times 10^{-36}$). Clindamycin targets the peptidyltransferase centre and inhibits protein synthesis by interfering with transfer RNA binding at the A-site. Bacteria have evolved enzymes that methylate the ribosome to prevent binding of several peptidyltransferase centre-binding antibiotics. LlmA only confers resistance to clindamycin (Table 1) and in fact increases sensitivity to pleuromutilins and linezolid (Supplementary Table 10).

**Resistance enzymes are variably shared among *Paenibacillus*.** We conducted a pan-*Paenibacillus* analysis of the resistance enzymes identified in this study to determine if any were recently acquired by *Paenibacillus* sp. LC231. To this end, we constructed a species tree of *Paenibacillus* using high-quality genome sequences (Supplementary Fig. 5). We then compared resistance enzymes among strains in a single clade containing *Paenibacillus* sp. LC231 (Fig. 5). Cpa, Mph, Vgb and Bah only appear in group 1, while Vat, Rph and Llm appear in all groups. Sequence diversity of enzymes did not always match with species diversity. We found that *Paenibacillus vortex* and *Paenibacillus* sp. FSL R5-808 are the most closely related strains to *Paenibacillus* sp. LC231, but their Vat enzymes are only 55% identical to VatI. In contrast, the Vat from *Paenibacillus lactis* is 94% identical to VatI. We analysed the genetic context of the *vat* genes in group 1 and found that neighbouring genes were not conserved; *vat* genes were found in four distinct contexts among the six strains (Fig. 5b). Furthermore, the *cpa* neighbouring genes are variable and *cpa* can even be lost (Fig. 5c). *P. lautus* ATCC 43898 is a closely related type strain (16S gene sequence is 99% identical to that of *Paenibacillus* sp. LC231). We generated a quantitative antibiogram for this organism that was very similar to that of *Paenibacillus* sp. LC231 from Lechuguilla Cave, demonstrating

that this multi-drug resistant phenotype is native to a clade within the *Paenibacillus* genus (Supplementary Table 2).

**Antibiotic-modifying enzymes are specific and efficient.** The pan-genome comparison of resistance enzymes revealed that, based on amino-acid sequence, they are conserved within closely related *Paenibacillus*, but they are also divergent from their characterized orthologues. We therefore performed steady-state kinetic analyses of each antibiotic-modifying enzyme (Supplementary Table 6). All were found to be effective catalysts at converting antibiotics into inactive products ($k_{cat}/K_m$ $10^4 - 10^5$). Aminoglycoside acetyltransferases can often modify a variety of aminoglycosides and we therefore evaluated the ability of AAC(2′)-IIb to modify a range of aminoglycosides. AAC(2′)-IIb specifically modified kasugamycin and no other aminoglycosides (Supplementary Fig. 6). AAC(2′)-IIa, a mobile resistance element in *Acidovorax avenae* ssp. *avenae,* is the only characterized homologue to AAC(2′)-IIb (ref. 21). The AAC(2′)-IIb $K_m$ for kasugamycin ($44 \pm 2 \,\mu\text{M}$) is an order of magnitude lower than for AAC(2′)-IIa ($475.3 \pm 141.2 \,\mu\text{M}$) and the turnover number ($k_{cat}$) for AAC(2′)-IIb ($2.95 \pm 0.03 \,\text{s}^{-1}$) is threefold higher than for AAC(2′)-IIa ($0.91 \pm 0.18 \,\text{s}^{-1}$).

**Discussion**

The origins of antibiotic resistance elements in the clinic have long been debated, but it is evident that non-pathogenic bacteria harbour numerous and diverse resistance genes orthologous and even identical to those emerging in pathogens. Here we show that careful exploration of the resistance genotype and phenotype of a single environmental organism can yield remarkable resistance diversity. This single organism is resistant to 26 of 40 antibiotics tested, and harbours five new resistance mechanisms and 12 orthologues of known resistance gene families. No resistance determinant from *Paenibacillus* sp. LC231 is found on mobile elements and therefore the risk of mobilization is low.

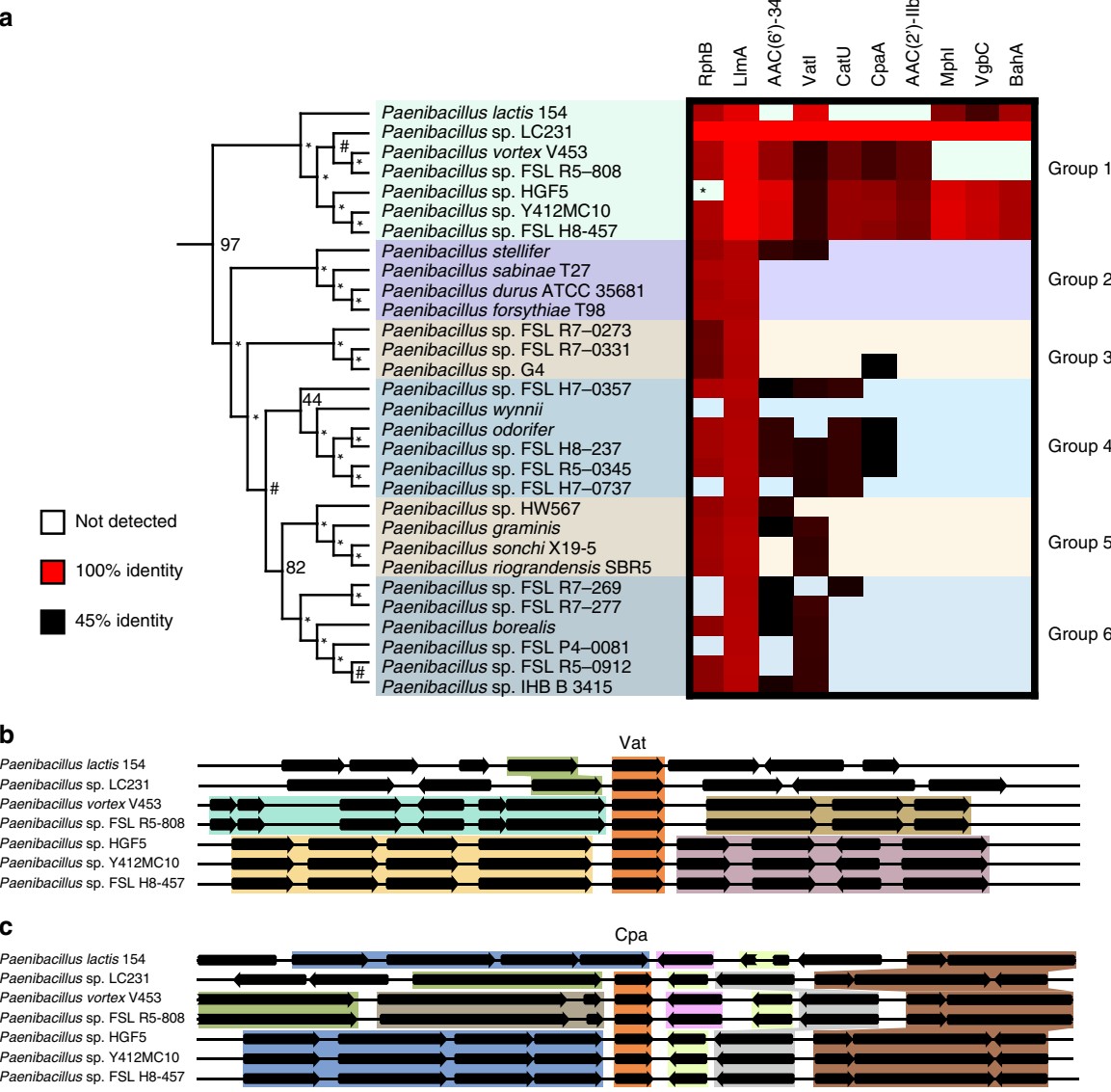

**Figure 5 | The conservation of resistance over millions of years. (a)** Ten resistance enzymes were compared across a single clade of the *Paenibacillus* species tree containing the closest surface relatives to *Paenibacillus* sp. LC231. The species tree is presented as a dendrogram. The presence or absence of each enzyme is indicated and the pair-wise sequence identity shown as a heat map. Background colour indicates that the enzyme was not detected. **(b,c)** The genomic location of *vat* (**b**) and *cpa* (**c**) genes across a subset of genomes reveals differing genomic context and synteny despite sequence conservation. An assembly gap in Rph from *Paenibacillus* sp. HGF5 is indicated with an asterisk.

Lechuguilla Cave is isolated from the surface and direct interaction of *Paenibacillus* sp. LC231 with pathogens is unlikely. However, our results demonstrate that surface *Paenibacillus* also have the same collection of resistance genes as LC231. More studies are needed to investigate the risk of genetic transfer from surface *Paenibacillus* to pathogens. The resistance genes in this study are functional in a heterologous host (*E. coli*) and could be selected for if combined with mobile elements. Our findings are consistent with metagenomics studies of waste water treatment plants[22], where most resistance genes are found native to the chromosome.

By combining genome sequence analysis, functional genomics and follow-up biochemical confirmation studies, we established a robust protocol to efficiently mine the intrinsic resistome of *Paenibacillus* sp. LC231. Our approach, which combines mining of genome sequences, functional genomics and rigorous biochemical study is widely applicable for dissecting bacterial resistomes, particularly for wild-type organisms without established genetic tools available for gene deletion studies. Furthermore, we show that even though this bacterium has been isolated from surface members of the same genus for millions of years, it retains a common multi-drug resistance genotype, although individual genes have diverged. While the resistance genes are not unique to *Paenibacillus* sp. LC231, the entire collection is not absolutely conserved among all closely related surface *Paenibacillus*.

These results speak to the pervasiveness and diversity of resistance in non-pathogenic environmental bacteria despite a lack of human intervention. This provides a reservoir of resistance elements that is substantial. The selective pressure of the misuse of antibiotics and the accumulation of residues in the environment should therefore be avoided lest these genes find opportunity to mobilize through bacterial populations, eventually making their way into pathogens.

## Methods

**Antibiotics and reagents.** All reagents were purchased from Sigma-Aldrich (Oakville, Ontario, Canada) unless otherwise specified. Acetyl-coenzyme A (CoA) was purchased from BioShop (Burlington, Ontario, Canada). Organic solvents were purchased from Fisher Scientific (Ottawa, Ontario, Canada). Telithromycin was purified from the pharmaceutical formulation Ketek (400 mg, Sanofi-Aventis US). Briefly, pills were crushed in a mortar and pestle, dissolved in acetonitrile at 40 °C, passed through an Amicon Ultra-15 10 kDa centrifugal filter unit and lyophilized. Linopristin and flopristin were gifts from AstraZeneca.

**Growth conditions and genomic DNA isolation.** *Paenibacillus* sp. LC231 was isolated from Lechuguilla Cave, as described[13]. To identify the potential for contamination by surface bacterial isolates, culture and preparation controls were used throughout sampling. Culture controls consisted of diluent and culture media mock-inoculated in the cave with sterile swabs, while preparation controls included all prepared media. All controls were incubated in the laboratory for 30 days under cave-relevant conditions (20 °C in the dark). None of the controls demonstrated any microbial growth, confirming that all observed colonies represented indigenous cave isolates. *P. lautus* ATCC 43898 was obtained from Cedarlane (Burlington, Canada). *Paenibacillus* strains were streaked from glycerol stocks (−80 °C) onto Tryptic Soy Agar and incubated at 30 °C for 2 days and cultures were re-streaked a second time before use. Several colonies with different growth phenotypes (that is, stationary and motile colonies) were always used for the experiments to ensure reproducibility. *Paenibacillus vortex* is a closely related strain that is also motile. Previous work in this strain has demonstrated that the stationary and motile colonies can have different antibiotic resistance phenotypes[23]. We found that using multiple colonies increased the reproducibility of our experiments. Liquid cultures were incubated with shaking (250 r.p.m.) at 30 °C for 2 days unless otherwise specified. The cell pellet from 3 ml of *Paenibacillus* sp. LC231 in tryptic soy broth (TSB) was used for genomic DNA isolation using a PureLink genomic DNA mini kit (Invitrogen). *E. coli* TOP10 (Invitrogen) was cultured in LB-Lennox and ampicillin (100 µg ml$^{-1}$) or kanamycin (50 µg ml$^{-1}$) were added to cultures harbouring plasmids. *S. aureus* RN4220 (gift from Dr Eric Brown, McMaster University) was cultured on LB agar, and *Staphylococcus saprophyticus* ATCC 15305 and a strain of *K. rhizophila* isolated from the soil by our lab (16s rRNA is 99% identical to *K. rhizophila* ATCC 9341) were cultured on Tryptic Soy Agar. *E. coli* strains were cultured in LB-Lennox (10 g typtone, 5 g yeast extract and 5 g NaCl).

**Genome sequencing and resistance prediction using the RGI.** The *Paenibacillus* sp. LC231 genome was sequenced using an Illumina MiSeq with the Reagent Kit v2 and paired-end 250 bp reads. Spades v3.5.0 was used to assembly the genome. The Resistance Gene Identifier (RGI) v3 (beta, accessed on 21 August 2015) was used to annotate resistance genes based on the curated CARD[15].

**Antimicrobial susceptibility and inactivation assays.** Antimicrobial susceptibility experiments followed Clinical and Laboratory Standards Institute (CLSI) guidelines for determining the MIC of compounds using the broth microdilution method[24] with the following modifications. For *Paenibacillus* strains, the saline suspension was made from several colonies with different growth phenotypes to ensure reproducibility. Ninety-six-well plates were incubated without shaking at 30 °C for 2 days (for *Paenibacillus* and *K. rhizophila*) or overnight at 37 °C (*S. aureus* and *E. coli*). Each MIC was performed in duplicate. Inactivation experiments for *Paenibacillus* sp. LC231 were set-up in the following way except for kasugamycin and capreomycin (see below). Antibiotics were added at $\frac{1}{4}$ MIC to 3 ml of Mueller Hinton Broth (MHB) and inoculated as described above but with shaking. Cells were removed by centrifugation at 21,000g for 5 min and the supernatant was assayed for antibiotic activity using a Kirby–Bauer disc diffusion assay and *K. rhizophila* or *S. saprophyticus* as the indicator organism. Inactivation was defined as a significant decrease in the zone-of-inhibition when compared with the sterile control.

Capreomycin and kasugamycin inactivation experiments were performed using *Paenibacillus* sp. LC231 lysate. A 250 ml culture in TSB was centrifuged at 10,000g and the cell pellet was washed with saline. The pellet was resuspended in 5 ml of 50 mM HEPES pH 7.5, 150 mM NaCl, 5 mM MgCl$_2$ and 1 mM β-mercaptoethanol, and cells were lysed at 30,000 psi using a One Shot cell disrupter (Constant Systems, Ltd.). The lysate was clarified by centrifuging at 18,000g for 20 min. Inactivation of capreomycin and kasugamycin was evaluated in a 50 µl reaction and the following conditions: 25 µl of clarified lysate in 50 mM HEPES pH 7.5, 150 mM NaCl, 0.1 mM EDTA and 0.5 mg ml$^{-1}$ of capreomycin or kasugamycin. The reactions were incubated at room temperature overnight and an equal volume of cold methanol was added. Samples were stored at −20 °C overnight and then centrifuged at 21,000g. Control experiments used 25 µl of clarified lysate that was boiled for 10 min. Antibiotic modification was evaluated using LC–MS. Reactions were analysed by injecting 10 µl onto an Agilent 1100 Series LC system and a QTRAP LC/MS/MS System (ABSciex). The reverse-phase HPLC conditions are as follows: isocratic 5% solvent B (0.05% formic acid in acetonitrile), 95% solvent A (0.05% formic acid in water) over 1 min, followed by

a linear gradient to 97% B over 7 min at a flow rate of 1 ml min$^{-1}$ and C18 column (Sunfire, 5 µm, 4.6 × 50 mm).

**Paenibacillus sp. LC231 library construction.** Genomic DNA (20 µg) was fragmented to 5 kb with a Couvaris S220 focused-ultrasonicator using red Minitubes (Covaris). Sheared DNA was end repaired with Fast DNA End Repair Kit (Thermo Scientific) according to the manufacturer's protocol. The End-Repair reaction (200 µl) contained 50 µl sheared DNA, 20 µl buffer and 10 µl enzyme mix. The repaired DNA was run on a 1% low-melting point agarose gel (1 × TAE) and fragments between 3 and 8 kb were excised using the GeneJET gel extraction kit (Thermo Scientific). Fragmented DNA was further purified using a GeneJET PCR purification kit (Thermo Scientific). Fragments were ligated into the SmaI restriction site of pUC19 in a 30 µl reaction (1.5 µl T4 DNA ligase (Thermo Scientific), 6 µl Fast Ligation buffer (Thermo Scientific), 120 ng SmaI digested pUC19 and 810 ng fragmented DNA) and incubated at room temperature for 2 h. *E. coli* ElectroMax DH10B electrocompetent cells (200 µl, Invitrogen) were transformed by electroporation with 5 µl of the ligation reaction and recovered in 9 ml of SOC media for 2 h at 37 °C and then plated evenly across 15 LB agar plates with ampicillin and incubated at 37 °C overnight. A measure of 5 ml of LB was added to each agar plate to facilitate scraping of *E. coli* clones. Resuspended cells from all plates were combined, washed in LB and resuspended in 25 ml LB with 15% glycerol and stored in 1 ml aliquots at −80 °C. Average fragment size was estimated by purifying plasmids from 12 clones and PCR amplifying the inserts using Phusion DNA polymerase (Thermo Scientific) with M13 forward and reverse sequencing primers.

**Selection of antibiotic resistant clones.** The *Paenibacillus* sp. LC231 genomic library in *E. coli* was plated at a density of 6–7 × the number of unique clones obtained in the transformation experiment. A measure of 3 µl of the library glycerol stock was diluted in 3 ml of warm LB and incubated with shaking for 1 h at 37 °C. Then, 200 µl was spread on pre-warmed LB agar plates with binary combinations of 100 µg ml$^{-1}$ ampicillin and one of the following antibiotics: 20 µg ml$^{-1}$ tetracycline, 50 µg ml$^{-1}$ kanamycin, 400 µg ml$^{-1}$ capreomycin, 400 µg ml$^{-1}$ kasugamycin, 500 µg ml$^{-1}$ tiamulin, 200 µg ml$^{-1}$ clindamycin and 100 µg ml$^{-1}$ mupirocin. Plates were incubated at 37 °C overnight with the exception of clindamycin, which was incubated an additional day at room temperature. Resistant clones were confirmed by replica plating onto media with 1–4 × the antibiotic concentration. Unique clones were identified using colony-PCR with M13 forward and reverse sequencing primers. Pure plasmids were submitted for Sanger sequencing (MOBIX, McMaster University, Canada). The results of Sanger sequencing were mapped to the assembled genome visualized in Geneious. The complete nucleic acid sequence of the insert was inferred from the genomic region between the mapped Sanger reads. Enzyme function of potential resistance enzymes (for example, acetyltransferase and hydrolase) was predicted by matching the protein sequence to a Pfam[25].

**Identification of BahA.** *E. coli* is naturally insensitive to bacitracin. The library glycerol stock described above was diluted 2 × 10$^{-6}$ in LB and 200 µl was spread on 35 LB agar plates containing ampicillin and incubated overnight at 37 °C to produce ∼300-well separated colonies. Each plate was replica plated onto two additional plates and incubated overnight at 37 °C. One plate, to be used to pick bacitracin-inactivating clones, contained ampicillin and was stored at 4 °C. The other plate lacked ampicillin and was overlaid with 5 ml of MHB with 0.75% agar and 4 µg ml$^{-1}$ zinc bacitracin. These plates were further incubated at 37 °C overnight to permit bacitracin inactivation by a clone expressing a bacitracin-modifying enzyme. An overnight culture of *K. rhizophila* was diluted one in three in sterile saline and spread on each plate using a cotton swab, which was then incubated at 30 °C for an additional two days. The bacitracin concentration used in this experiment is 2 × the MIC of *K. rhizophila* under these conditions and completely inhibits colony formation. *E. coli* clones with a halo of yellow *K. rhizophila* growth were identified as expressing a bacitracin-inactivating enzyme. To confirm this result, the corresponding colonies from the replica plate were used to inoculate 3 ml of LB with 200 µg ml$^{-1}$ zinc bacitracin. BahA was identified by Sanger sequencing.

**Construction of a Paenibacillus species tree.** A *Paenibacillus* species tree was generated using 10 housekeeping genes; *atpD, dnaA, gyrB, pgi, pyrH, recA, rpoB, sucC, topA* and *trpB*[26]. Sequenced *Paenibacillus* strains with full sequences for all 10 housekeeping genes were used in phylogenetic analysis, except *Paenibacillus vortex*, which did not have an intact *topA* sequence (Supplementary Table 11)[27]. *Bacillus megaterium* DSM319 (CP001982.1) and *Brevibacillus brevis* NBRC 100599 (AP008955.1) were used as an outgroup. Each housekeeping gene was aligned with MAFFT (L-INS-i)[28], trimmed with TrimAl v. 1.4 rev15 using the automated1 setting[29], and concatenated. A maximum-likelihood tree was generated using RAxML with rapid Bootstrap analysis on 1,000 replicates. 'X's were used in place of the missing *Paenibacillus topA* sequence. All genome sequences were accessed on 10 July 2015.

**Cloning of resistance genes and susceptibility testing.** Cloning is summarized in Supplementary Table 12. TaeA and TetAB(48) are ABC-transporters composed of one and two polypeptides, respectively. Primers were designed to amplify 500 bp upstream of *tetAB(48)* so to clone both genes with their native promoter in pUC19. Similarly, *taeA* was cloned into pUC19 with the 500 bp upstream region. *E. coli* TOP10 expressing either TetAB(48) or TaeA were used in MIC experiments. The C-terminal domain of *bahA* (corresponding to residues 200–756) was first cloned into pET11a, then PCR-amplified with the upstream ribosome-binding site and XbaI-restriction site and cloned into pET21a, which includes an in frame C-terminal His$_6$-tag. The pET11a-*bahA* construct was transformed into *E. coli* BL21(DE3) for MIC experiments. In bacitracin inactivation experiments, 3 ml of LB containing 200 μg ml$^{-1}$ bacitracin was inoculated with either *E. coli* BL21(DE3) pET11a or *E. coli* BL21(DE3) pET11a-*bahA* and cultured with shaking overnight. Inactivation was assayed using a Kirby–Bauer assay and using LC–electrospray ionization (ESI)–MS as described above. The pET21a constructs of *aac(2')-IIb* and *aac(6')-34* were used for MIC experiments. All pET vectors were transformed into *E. coli* BL21(DE3) for MIC experiments.

**Pan-*Paenibacillus* resistance enzyme analysis.** Orthologues of ten resistance enzymes from *Paenibacillus* sp. LC231 (RphB, VatI, AAC(2')-IIb, AAC(6')-34, MphI, CatU, VgbC, LlmA, CpaA and BahA) were identified in a subset of *Paenibacillus* genomes using an extended BLASTp method. Briefly, each protein sequence (for example, RphB and VatI) was queried against GenBank limited to txid44249 (*Paenibacillus*) using an *e*-value cutoff of $1 \times 10^{-10}$. Enzymes that were at least 50% identical over at least 80% of the seed sequence were in turn queried against GenBank to identify more diverse *Paenibacillus* orthologues using the same BLAST parameters. The pair-wise sequence identity of each enzyme with its orthologue in *Paenibacillus* sp. LC231 was computed with Clustal Omega[30] per cent identity matrix and plotted as a colour gradient from 45% (the most diverse orthologue) in black to 100% in red. For genomic context analysis, sequence and annotations for 5 kb upstream and downstream of each gene were extracted from GenBank files using RefSeq annotations. Arrows representing genes were made in Geneious. Genes from *Paenibacillus* sp. LC231 were manually annotated using BLASTx.

**Purification of antibiotic inactivating enzymes.** Antibiotic-inactivating enzymes were overexpressed in *E. coli* BL21(DE3) for protein overexpression. A 3 ml overnight culture was used to inoculate 1 l of LB and incubated with shaking at 37 °C with either kanamycin (pET28a) or ampicillin (pET11a, pET21a or pET22b) until an OD$_{600}$ of 0.6–0.8, at which point the cultures were placed in an ice bath for 20 min and protein expression was induced with 1 mM isopropyl-β-D-thiogalactopyranoside at 16 °C overnight. Cells were collected at 5,000*g* for 20 min and washed in cold saline. The cell pellet was resuspended in 20 ml of 50 mM HEPES pH 7.5, 150 mM NaCl, 5% glycerol and 5 mM imidazole (buffer A) and lysed using a One Shot cell disrupter (Constant Systems, Ltd.) at 20,000 psi. A measure of 5 mg of bovine pancreas DNase, 2.5 mg of bovine pancreas RNase and an additional 15 ml of buffer were added and centrifuged at 40,000*g* for 45 min. Overexpressed proteins were purified using ion-metal affinity chromatography (1 ml Ni$^{2+}$-nitrilotriacetic acid column, Qiagen) equilibrated with buffer A. A linear gradient was used to elute protein over 20 column volumes starting at 94:6 buffer A: buffer B to 100% buffer B (50 mM HEPES pH 7.5, 150 mM NaCl, 5% glycerol and 250 mM imidazole) on an ÄKTA purifier (GE Scientific). Fractions containing pure protein were identified using SDS–polyacrylamide gel electrophoresis, pooled and desalted with 50 mM HEPES pH 7.5 using a PD-10 gel filtration column (GE), except for CatU and VatI where the buffer contained 150 mM NaCl. In addition, 0.1 mM β-mercaptoethanol was added to purified CatU. Pure enzyme stocks were stored at 4 °C.

**MS of antibiotics inactivated by purified enzymes.** The predicted catalytic functions of CpaA, VatI, MphI, CatU, VgbC, RphB, AAC(2')-IIb and AAC(6')-34 were confirmed with LC–MS analysis of enzyme reactions. Each 250 μl reaction consisted of 1 mg ml$^{-1}$ antibiotic, 60–90 μg ml$^{-1}$ of enzyme and 1× reaction buffer (see below). Specific enzyme/substrate combinations were as follows; CpaA/capreomycin, VatI/flopristin, MphI/telithromycin, CatU/chloramphenicol, VgbC/linopristin, RphB/rifampin, AAC(2')-IIb/kasugamycin and AAC(6')-34/ sisomicin. The reaction for CpaA, VatI, CatU and AAC(2')-IIb contained 50 mM HEPES pH 7.5, 150 mM NaCl, 0.1 mM EDTA and 1 mM acetyl-CoA. The MphI reaction contained 50 mM HEPES pH 7.5, 40 mM KCl, 10 mM MgCl$_2$ and 1 mM GTP. The reaction for VgbC contained 50 mM HEPES pH 7.5 and 1 mM MgCl$_2$. The reaction for RphB contained 50 mM HEPES pH 7.5, 40 mM NH$_4$Cl, 5 mM MgCl$_2$ and 2.5 mM ATP. The AAC(6')-34 reaction contained 25 mM MES pH 6.0, 1 mM EDTA and 1 mM acetyl-CoA. Reactions were incubated at room temperature overnight. An equal volume of cold methanol was added, stored at −20 °C overnight and centrifuged at 21,000*g* for 10 min. A volume of 10–20 μl of each reaction was injected onto an Agilent 1100 Series LC system and a QTRAP LC/MS/MS System using the HPLC conditions described in online methods.

**Structural characterization of inactivated bacitracin.** A measure of 0.5 ml of a 3 ml overnight *Paenibacillus* sp. LC231 culture in MHB was used to inoculate 50 ml of MHB. The culture was centrifuged at 4,000*g* for 40 min at room temperature, washed with 10 ml of sterile saline and resuspended in 1.5 ml MHB. A volume of 750 μl of this suspension was boiled for 10 min as a negative control. Bacitracin A (2.5 mg ml$^{-1}$) was added to the suspension and incubated overnight with shaking at 30 °C. Cells were removed by centrifuging at 21,000*g*. An equal volume of methanol was added, stored at −20 °C overnight and centrifuged at 21,000*g* for 10 min. LC–ESI–MS$^n$ experiments were performed on a ThermoFisher LTQ-XL-Orbitrap Hybrid mass spectrometer (ThermoFisher, Bremen, Germany) in positive-ion mode. A volume of 20 μl of the reaction or bacitracin A (1 mg ml$^{-1}$) were injected onto an Agilent 1290 Infinity UPLC system using the following chromatography conditions: isocratic 5% solvent B (0.05% formic acid in acetonitrile), 95% solvent A (0.05% formic acid in water) over 1 min, followed by a linear gradient to 100% B over 4.5 min at a flow rate of 0.4 ml min$^{-1}$ and C18 column (info needed). The conditions for ESI–MS$^n$ experiments are as follows: sheath gas flow rate at 40, auxiliary gas flow rate at 10, ion spray voltage at 4.2 kV, capillary temperature at 340 °C, capillary voltage at 29 V, tube lens voltage at 120 V and normalized collision energy at 27%. MS$^1$ was set to fourier transform (FT) full scan between 500–1,700 *m/z* with a resolution set at 60,000 followed by MS$^2$ and MS$^3$ of the most intense ions from MS$^1$ and MS$^2$ stages, respectively. Ion fragments in MS$^3$ were compared with previous reports (Supplementary Table 5)[31].

**CpaA characterization.** CpaA substrate specificity was examined by monitoring CoA liberation with 4,4′-dithiodipyridine (DTDP) at 324 nm. Reactions were performed at 25 °C in a 96-well plates with a final volume of 250 μM and contained 50 mM HEPES pH 7.5, 150 mM NaCl, 0.1 mM EDTA, 1.7 mM DTDP, 250 μM acetyl-CoA and 5.6 μg ml$^{-1}$ CpaA. Reactions were initiated with the addition of 100 μM capreomycin, kanamycin or viomycin and monitored for 1,000 s. The site of acetylation was determined using NMR. A large-scale reaction consisted of 25 mM ammonium bicarbonate pH 7.8, 12 mg capreomycin (mixture of IA and IB), 15 mM acetyl-CoA, and 14 μg CpaA in a final volume of 1 ml and was incubated at 25 °C overnight. An additional 10 mM acetyl-CoA and 28 μg CpaA was added, and the reaction was incubated for 8 h. The solution was lyophilized, dissolved in 0.5 ml of 5 mM ammonium bicarbonate and the pH was adjusted to 9.0 using ammonium hydroxide. The sample was purified by anion-exchange chromatography (1 ml Q-sepharose HiTrap XL) and the compound was eluted in 5 mM ammonium bicarbonate pH 9.0. The eluate was lyophilized and the compound was further purified using normal-phase flash chromatography (12 g RediSep Rf silica, Teledyne) and eluted with butanol:acetic acid:water using a linear gradient from 3:1:1 to 3:3:4 to yield 10 mg acetyl-capreomycin. The compound was dissolved in 500 μl D$_2$O and 1 μl glacial acetic acid for one-dimensional and two-dimensional NMR experiments.

The molecular formula of 1-N-acetyl-capreomycin IA was determined as C$_{27}$H$_{46}$N$_{14}$O$_9$ according to its positive HRESIMS at *m/z* 711.3633 [M + H]$^+$. Careful comparison of the $^1$H-NMR data of 1-N-acetyl-capreomycin IA with those of capreomycin IA showed that H-1 in 1-N-acetyl-capreomycin IA shifted to downfield 0.23 ppm, H-2a and H-2b shifted to upfield 0.20 and 0.24 p.p.m., respectively (Supplementary Fig. 3; Supplementary Tables 7 and 8). In the $^{13}$C-NMR spectrum, C-16 shifted to downfield 3.9 p.p.m., indicating that a acetyl group connected to N-1. This connection was further confirmed by key heteronuclear multiple-bond correlation spectroscopy correlations from H-1 (at $\delta_H$ 4.60 p.p.m.) to CH$_3$C=O (at $\delta_C$ 174.0 p.p.m.).

**MphI characterization.** MphI substrate specificity was examined using a coupled assay[32]. Reactions were performed in a 96-well plate with a final volume of 250 μl and contained 50 mM HEPES pH 7.5, 40 mM KCl, 10 mM MgCl$_2$, 0.3 mM NADH, 3.5 mM PEP, 4.8 U PK/LDH, 1 mM GTP and 800 μM macrolide (erythromycin, clarithromycin, azithromycin, tylosin, telithromycin, spiramycin and roxithromycin). The assay plate was incubated at 37 °C for 5 min and reactions were initiated by the addition of GTP. Acid hydrolysis was used to remove the C3-cladinose from clarithromycin. Clarithromycin (19.7 mg, 26.3 mmol) was added in several portions over 5 min to a mixture of 500 μl water and 50 μl concentrated HCl. The reaction proceeded at room temperature, with stirring, for 2 h. The crude material was then purified using reverse-phase flash chromatography (C18, 5.5 g RediSep Rf Gold column, Teledyne) in a linear gradient of 100% water to 60:40 water:acetonitrile over 10 min. The fractions containing pure compound, as determined by LC–MS, were pooled and lyophilized. The compound structure was confirmed to be descladinose clarithromycin using one-dimensional and two-dimensional NMR experiments (Supplementary Figs 4 and 5; Supplementary Table 9). The yield was 58.4%. Descladinose clarithromycin was used in an MphI enzyme assay without PK:LDH in the following reaction; 0.5 mg ml$^{-1}$ of descladinose clarithromycin, 22.5 μg MphI and 1 mM GTP in a final volume of 100 μl. A 50 μl aliquot was removed immediately after mixing and the reaction was stopped with an equal volume of cold methanol. The remainder of the reaction was incubated at 37 °C for 16 h. A negative control with clarithromycin was used for comparison. Each sample was analysed by LC–MS.

**Kinetic characterization of antibiotic inactivating enzymes.** Enzyme kinetics experiments were performed in triplicate except where noted. All reactions were initiated by adding enzyme except for MphI, which was initiated by adding GTP.

All enzyme reactions were performed in 96-well Nunc plates (Thermo Scientific) in a Spectramax Plus384 (Molecular Devices) microtitre plate reader. GraphPad Prism was used for data analysis.

Steady-state kinetics of RphB was performed at 25 °C by monitoring inorganic phosphate production using the EnzChek Phosphate Assay Kit (Molecular Probes) in 50 mM HEPES pH 7.5, 40 mM NH$_4$Cl, 5 mM MgCl$_2$ with a final volume of 100 μl[16]. The $K_m$ of ATP was determined as follows; 25 μM rifampin, 7.81–500 μM ATP and 68.5 nM RphB. The $K_m$ of rifampin was determined as follows; 1 mM ATP, 0.225–25 μM rifampin and 68.5 nM RphB. Enzyme reactions were performed in duplicate.

Steady-state kinetics of MphI was performed at 25 °C and in 50 mM HEPES pH 7.5, 40 mM KCl, 10 mM MgCl$_2$ in a volume of 250 μl. The $K_m$ of GTP was determined as follows: 400 μM tylosin, 3.12–2,000 μM GTP and 430 nM MphI. The $K_m$ of macrolides were determined as follows: 200 μM GTP, 2.34–800 μM macrolide and 430 nM MphI.

Steady-state kinetics of VgbC was performed by monitoring the decrease in absorbance associated with the linearization of streptogramin B antibiotics[33]. Enzyme reactions were performed at 25 °C and in 50 mM HEPES pH 7.5 and 1 mM MgCl$_2$ in a volume of 250 μl. Linopristin was used as the substrate and the decrease in absorbance was monitored at 305 nm ($\varepsilon = 6{,}220\,M^{-1}cm^{-1}$). For the $K_m$ of linopristin: 2.34–200 μM linopristin and 29.6 nM VgbC.

Steady-state kinetics of CatU was performed by monitoring CoA liberation using 5,5′-dithiobis-(2-nitrobenzoic acid) (DTNB) at 25 °C (ref. 34). Reactions were performed in 50 mM HEPES pH 7.5, 150 mM NaCl, 0.1 mM EDTA and 1 mM DTNB in a final volume of 200 μl. The $K_m$ of acetyl-CoA was performed as follow: 400 μM chloramphenicol, 31.3–2,000 μM acetyl-CoA and 62.5 nM CatU. The $K_m$ of chloramphenicol was performed as follows: 500 μM acetyl-CoA, 4.69–500 μM chloramphenicol and 62.5 nM CatU. Acetyl-CoA was not saturating under these conditions.

Steady-state kinetics of VatI were performed in a volume of 250 μl at 25 °C and in 50 mM HEPES pH 7.5, 150 mM NaCl, 0.1 mM EDTA[35]. The $K_m$ of acetyl-CoA was performed as follows: 100 μM flopristin, 7.81–1,000 μM and 13.8 nM VatI. The $K_m$ of flopristin was performed as follows: 250 μM acetyl-CoA, 3.91–500 μM flopristin and 13.8 nM VatI.

Steady-state kinetics of AAC(6′)-34 was performed by monitoring CoA liberation using DTDP in a 250 μl reaction at 25 °C containing 25 mM MES pH 6.0, 1 mM EDTA[36]. The $K_m$ of acetyl-CoA was performed as follows: 35 μM sisomicin, 18.8–1,000 μM acetyl-CoA and 120 nM AAC(6′)-34. The $K_m$ of sisomicin was determined as follows: 500 μM acetyl-CoA, 1.40–250 μM sisomicin and 120 nM AAC(6′)-34.

Steady-state kinetics of AAC(2′)-IIb were performed similar to AAC(6′)-34, but in 50 mM HEPES pH 7.5, 150 mM NaCl, 0.1 mM EDTA in a volume of 250 μl (ref. 21). The $K_m$ of acetyl-CoA was performed as follows: 250 μM kasugamycin, 1.56–1,000 μM acetyl-CoA and 21.2 nM AAC(2′)-IIb. The $K_m$ of kasugamycin was performed as follows: 125 μM acetyl-CoA, 7.81–1,000 μM and 21.2 nM AAC(2′)-IIb. A similar assay was employed to examine the substrate specificity of AAC(2′)-IIb; 250 μM acetyl-CoA, 21.2 nM AAC(2′)-IIb and 50 μM of either kasugamycin, apramycin, ribostamycin, fortimicin A, gentamicin, paromomycin, lividomycin, amikacin or neomycin.

CpaA steady-state kinetics was performed at room temperature in 250 μl and in 50 mM HEPES pH 7.5, 150 mM NaCl, 0.1 mM EDTA. The $K_m$ of acetyl-CoA was performed as follows: 5 μM capreomycin, 5.63–500 μM acetyl-CoA and 37.5 nM CpaA. The $K_m$ of capreomycin was performed as follows: 250 μM acetyl-CoA, 0.19–10 μM capreomycin and 37.5 nM CpaA.

**Data availability.** Nucleotide sequences for the *Paenibacillus* sp. LC231 genome have been deposited in the GenBank WGS database with the accession code JFOM00000000. The resistance determinants studied here were deposited in the GenBank Nucleotide database with accession codes KX531043 to KX531056. In addition, each resistance determinant was deposited in CARD (https://card.mcmaster.ca/). The authors declare that all other data supporting the findings of the study are included in this published article and its Supplementary Information files, or are available from the corresponding author on request.

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

## Acknowledgements

We thank members of the CARD team for input into antibiotic resistance sequence analysis. We also thank Nicholas Waglechner (McMaster University) for helpful

discussions in generating the *Paenibacillus* species tree and Christine King (McMaster University) for genome sequencing and shearing genomic DNA. This research was funded by a Canadian Institutes of Health Research Grant (MT-13536), Natural Sciences and Engineering Research Council Grant (237480) and by a Canada Research Chair in Antibiotic Biochemistry (to G.D.W). A.G.M. holds a Cisco Research Chair in Bioinformatics, supported by Cisco Systems Canada, Inc.

## Author contributions

A.C.P. performed all experiments except for structural elucidation using NMR. W.W. assisted in acetyl-capreomycin purification and performed acetyl-capreomycin NMR experiments. K.K. assisted in purifying descladinose clarithromycin and structural characterization of inactivated bacitracin, and performed descladinose clarithromycin NMR experiments. A.G.M. assisted the antibiotic resistance sequence analysis. H.A.B. collected strains and provided growth parameters. A.C.P and G.D.W. designed research and wrote the paper with contributions from all authors.

## Additional information

**Competing financial interests:** The authors declare no competing financial interests.

