## [Peer Review File · Nature Communications]

Reviewers' comments:

Reviewer #1 (Remarks to the Author):

The study by Pawlowksi et al describes several new types of resistance/tolerance mechanisms in an environmental bacterium from an isolated cave. Resistance/tolerance mechanisms were identified both by sequence comparisons to known resistance genes, and by functional genomics through expression of gene fragments in *E. coli*, followed by biochemical characterization of the identified enzymes. This, particularly the biochemical characterization, is well performed and together serves as a good example of how to explore the resistome of a bacterium (although the authors could elaborate more on alternative/complementary approaches and their respective benefits and disadvantages). Because of the site where the bacterium was isolated (isolated cave), and based on comparative genome analyses, it is correctly concluded that the mechanisms identified are of ancient origin. From this, the authors point out that there is likely a large reservoir of unknown tolerance/resistance mechanisms in the environment, and stress the risk if such genes would be transferred to human pathogens, particularly under the exposure to antibiotic residues. This is perhaps not a really surprising or novel conclusion, but deserves to be stressed and exemplified and backed up more in the literature, as is done here in a nice way. Risks for such transfer events (based on genetic context and other factors) are not really discussed in the manuscript, although functionality in *E. coli*, as demonstrated for several genes, adds to the perceived risk. This observation could be mentioned and discussed together with a range of other factors that influence the risk for gene transfer and eventually treatment failure.

The authors speculate on the presence of antibiotic producers in this type of environment, based on a high multi-resistance profile. However, when they compare the resistance profile of the investigated species to the profile of one single related species isolated from another type of environment, the two strains were in fact relatively similar (see supplementary tables 1 and 2). This really does not stress the cave environment as an environment where bacteria are particularly "resistant" (also see below). Hence, the conclusion that the cave environment would be particularly interesting for exploring antibiotic producers is not really substantiated to any large extent by the findings presented in this manuscript. A comparison of resistance or tolerance patterns in a range of bacterial communities from different types of environments, assessed with comparable methodology, would be needed to support such a conclusion. Overall, the "cave" component of the study basically only serves the purpose of providing evidence that the observed "resistance" is intrinsic and not the result from recent changes or human activities.

The authors are unclear about the resistance concept, i.e. they do not define what they mean with "resistance". In the broader literature, resistance is usually referred in either one of two ways: The most common one is defined from a clinical standpoint, that is a level of tolerance to an antibiotic that makes it difficult or impossible to treat an infection with the given antibiotic. This is the basis for

the cut-off values for resistance for specific pathogens. What level is defined as resistant from this clinical definition varies between species. For the genus investigated here (*Paenibacillus*), there is no clinical cut-off values to my best knowledge. The other definition commonly used is an ecological one, based on an increased tolerance of a bacterial strain in comparison with another strain of the same species, regardless of the level of increase or the absolute level of antibiotic required to completely stop growth. This is a definition applicable to any cultivable species, but it focus on "acquired" resistance, which is not really the case here. It therefore becomes a bit difficult to use the term resistance in this context where "intrinsic resistance" is studied in a non-pathogenic species. In the author's previous study on bacteria from the same cave, they used a "home-made" definition, with a cutoff of 20 mg/L for all antibiotics (i.e. not really comparable to other studies). For some antibiotics, that was considerably below the clinical breakpoints for most pathogens, whereas it was higher for others. In the present paper, the authors report the actual MIC values for each antibiotic, which is considerably more informative. On line 95 they state that "26 had an MIC at least 8x higher than susceptible strains", but it is unclear what strains they are talking about. Maybe they mean some other clinically important species? In the reviewer's opinion, the terminology and use of the resistance concept should be clarified and more stringent.

Some genes discovered by functional genomics were not discovered by similarity searches to the CARD database. It should be stressed how dissimilar from the closest resistance genes present in CARD those genes were, and the similarity search strategy applied should be explained.

Reviewer #2 (Remarks to the Author):

This manuscript (MS) is a very thorough analysis of the resistome of one bacterial strain isolated from a cave which had not been exposed for over 4 million years. There are always concerns over contamination in such cases but I am convinced that sufficient controls were done and are reported in the MS. What is exceptional about this one strain of the genus *Paenibacillus* is the 18 different resistance mechanisms which were defined in terms of specific chromosomal genes all cloned and expressed in *E. coli*. No mention is made of mobility so presumably these genes are not readily mobilised? The biochemical analysis of each resistance mechanism has been exemplary and the detail is extremely comprehensive. The authors acknowledge that resistance is an ancient phenomenon which has been discovered in many bacteria from the pre-antibiotic era so these observations in an environmental bacterium are not new. What is new is the description and detail of mechanisms in such an isolate irrefutably isolated for over 4 million years. Although it is now becoming clear from other published work that there is a wealth of resistance in environmental bacteria distinct from those mechanisms well defined in clinically important bacterial. An excellent

example of this is the report in Nat. Comms by Munck et al., 2016 who cloned resistance genes from metagenomic DNA isolated from the bacterial biomass of WWTP digestors. Less than 10% of resistance genes were shared in other metagenomes and many were highly diverse thus supporting the huge reservoir of resistance. They (Munck et al.) did not observe any significant evidence of gene mobilisation except for spectinomycin gene which was mobilised by class 1 integron. I think this may correlate with use of spectinomycin in agriculture. What would be useful to know is how the genes can be mobilised and under what kind of selection if any. Clearly the continued extensive use of antibiotics in agriculture and further dissemination into the environment may act as a catalyst for gene mobilisation. This point is well made and some more comment about mobilisation would be helpful.

Reviewer #3 (Remarks to the Author):

In this paper, Pawlowski et al perform an extensive analysis of the antibiotic resistance phenotype of *Paenibacillus* sp. LC231, a strain that was isolated from Lechuguilla Cave, which forms an underground ecosystem that has been isolated for millions of years. Through an elegant combination of microbiology, microbial genomics and chemistry, the authors show that the *Paenibacillus* strains carries 18 antibiotic resistance determinants (including 5 new enzyme families and 3 novel resistance determinants). These data show that bacteria from pristine environments can be rich in resistance genes, providing evidence (beyond other publications of the Wright group and others) that the presence of antibiotic resistance genes are natural phenomena. There is no evidence that the 'ancient' cave strain of *Paenibacillus* is, in any way, remarkable as *Paenibacillus* strains from other (non-isolated) sites can have very similar antibiotic resistance profiles.

While I have no reason to doubt the validity of the approach or the quality of the data, I believe some aspects (mainly in terms of the presentation of the data) will need to be improved or corrected. These are specifically outlined below.

There are no issues with the statistical analysis of the data in this manuscript and references are appropriate. The manuscript is generally well-written but contains a few errors in language or style (outlined below).

The conclusions (particularly for the resistance determinants that were biochemically characterized) of resistance determinants of *Paenibacillus* LC231 are supported by the data. However, I do not agree with the statement that the data indicate that 'new antibiotic producers may be present in Lechuguilla Cave' (l. 35). I fail to see why Lechuguilla Cave should contain new antibiotic producers and why it is singled out as a site 'worthy of exploration for antibiotic leads' (l. 244). While it is certainly possible that new antibiotic producers can be found in the cave, there is no reason to believe that the cave is a more promising environment than other sites (soil, water etc) to find antibiotic producers. I feel these statements should be removed and the conclusions should focus on the in-depth characterization of the resistance gene repertoire of *Paenibacillus* LC231.

Suggested improvements:

I. 34. 'Longevity of these genes in this environment'. I believe the authors refer to the cave system with 'this environment' but the data point towards a natural reservoir for resistance genes in *Paenibacillus* inside or outside the cave. Therefore 'this environment' should be replaced by 'this genus' or this phrase should be removed.

I. 61. Combinatorialization: why not simply 'combination of multiple genes on a single genetic platform' or similar?

I. 94 - 98. This section is somewhat confusing. Subjective terms like 'highly resistant' should be avoided and data should be described in a more objective, quantitative fashion. Antibiotics do not have an MIC ('26 had an MIC at least 8x higher', which is a difficult-to-understand phrase anyway) but strains have an MIC for specific antibiotics. I believe the comparison with 'susceptible strains' is done with *M. luteus* and *S. aureus* but these data are not included in the manuscript, so it is not possible to check for which antibiotics the MICs are {greater than or equal to}8-fold higher. I suggest Supplementary tables 1 and 2 are to be combined in a single table and the data for *S. aureus* and *M. luteus* are added. The statement on inactivation of antibiotics is, at this point in the manuscript, not supported by the data.

I. 99 - 115. In this section (putative) resistance genes are detected using genome sequencing followed by interrogation of the CARD database. The correlations between resistant phenotypes and detection of resistance determinants are not always convincing. The authors mention 'decreased rifampin sensitivity' but the MIC for this antibiotic is only 1 µg/ml, which is generally considered sensitive. The role of *rph* has not been substantiated so I. 104 should read 'may inactivate'. 'These drugs' (I. 105) should be corrected to 'this drug'. The role of the *Cfr*-like gene is also not clear. How can the authors be so sure that it is responsible for the slightly decreased sensitivity to a wide range of antibiotics from different classes. As with rifampin, the resistance phenotypes are not at all clear (e.g. MIC for linezolid of LC231 is 1 µg/ml, which would generally be considered susceptible; clindamycin resistance is linked to *cfr* in supplementary table 1, but is not discussed in text). The gene appears not have been heterologously expressed in *E. coli* (Table 1). I believe these experiments should be performed to provide additional evidence for the role of the *cfr* gene in LC231. Finally, as the authors have a complete genome sequence, I feel they should add information on the location (chromosome, plasmid) and/or genetic context of the resistance gene as it is interesting to know whether resistance genes appear to be fixed on the chromosome or are carried on mobile genetic elements. Genome sequence data should be submitted to NCBI or ENA and an accession number should be provided in the manuscript.

I. 131 - 132: I note that not all 'new resistance elements' are discussed in the manuscript (e.g. TetAB(48) and TaeE) so this line should be rephrased.

I. 154, I. 190. Add 'e-value: ' to the score for clarity

I. 177 - 178. Please add quantitative information on the relatedness of MphI to other macrolide kinases.

I. 193 - 194: 'confer' should be corrected to 'confers'. What is the mechanism for the increased sensitivity to linezolid and pleuromutilins? *E. coli* is intrinsically resistant to these antibiotics.

I. 201. Suggest to replace 'track with' with 'match'

I. 204. It is not entirely clear to what gene/organism the Vat protein from *P. lactis* is 94% identical.

I. 208. *P. lautus* is not present in Fig. 5, presumably because no genome sequence is publicly available. However, it is of considerable interest to determine the presence of resistance genes of this organism as well, as it appears closely related to LC231 (in addition, please write 16S rRNA gene sequence in I. 208). As this strain is modern (isolated from the intestinal tract of a child), it shows that LC231 is not particularly different from currently circulating *Paenibacillus* strains.

I. 211. It is perhaps better to write 'a clade within the genus *Paenibacillus*' to highlight that not all *paenibacilli* share the multi-drug resistant phenotype (see Fig 5A).

I. 242. What is meant with 'resource competitive'?

I. 248. Delete 'remarkable and'

I. 249. I am not sure if one can claim that the environmental reservoir of resistance is 'underappreciated'.

I. 249 - 252. Perhaps the authors can shortly discuss the likelihood of the resistance genes from LC231 being mobilized to pathogens (e.g. are they located on plasmids? See comments above). In addition, it could be said that these resistance genes are unlikely to be particularly dangerous, even if they end up in opportunistic pathogens like *Staphylococcus aureus* and *Klebsiella pneumoniae* as most appear to have a remarkably narrow substrate range and most do not appear to target clinically important antibiotics: this could also be discussed here.

I. 288: Cedarlane: add city, country.

I. 297: What is the strain name of the *M. luteus* isolate used in this study? If it is ATCC 9341, it should be renamed as *Kocuria rhizophila* (see Tang & Gilbert. *Int J Syst Evol Microbiol* 2003 53:995)

I. 300 - 301: the reference should be put after 'broth microdilution method' and correct 'modification' to 'modifications'.

I. 306. The abbreviation MHB (for Muller Hinton Broth) is not explained

I. 347. Write 'strains' or 'isolates' after *Paenibacillus*.

Fig 2: The active version of bacitracin is the X = -NH₂ version. Please correct.

Fig 3: The panels of this figure have been switched. Terms like multi-bond HMBC, TOCSY and HSQC correlations are not explained in the manuscript and are difficult to follow to readers without expertise in NMR.

Fig 5. I assume the dendrogram-type tree a representation of the phylogenetic tree in Fig 6? If yes, please specify this in the legend. Please write 'An assembly gap in *RphB* is indicated with an asterisk'.

Table 1: the MICs of the wild-type strain and the strains expressing the resistance determinants should be shown (rather than a fold change in MIC). I note that a fold change of 0 (for mphI and erythromycin) is most likely a mistake (no change in resistance is a fold change of 1).

Supplementary information:

l. 29 - 30. I believe this figure can be deleted if my comments on Table S1 and S2 are addressed. It is also not clear what the authors mean with 'a value equal-to or greater-than is indicated': equal-to, greater-than what?

What are 1x Cpa buffer and 1x Mph buffer: presumably the buffer described in l. 267 and l. 268? Please specify. For the experiments in l. 326 - 373 it is also not clear in which buffers these experiments were performed (presumably the same as outlined earlier, e.g. l. 267 - 271?): this should be specified. In l. 326 - 373, the phrase 'Km was determined as follows' followed by the concentration of reagents is insufficient to reproduce the experiment and should be rewritten.

Reviewer #1:

*The study by Pawlowksi et al describes several new types of resistance/tolerance mechanisms in an environmental bacterium from an isolated cave. Resistance/tolerance mechanisms were identified both by sequence comparisons to known resistance genes, and by functional genomics through expression of gene fragments in E. coli, followed by biochemical characterization of the identified enzymes. **This, particularly the biochemical characterization, is well performed and together serves as a good example of how to explore the resistome of a bacterium** (although the authors could elaborate more on alternative/complementary approaches and their respective benefits and disadvantages).*

Many thanks! We have commented on the strength of our approach versus others on lines 245-248. (Our approach, which combines mining of genome sequences, functional genomics and rigorous biochemical study is widely applicable for dissecting bacterial resistomes, particularly for wild-type organisms without established genetic tools available for gene deletion studies)

Because of the site where the bacterium was isolated (isolated cave), and based on comparative genome analyses, it is correctly concluded that the mechanisms identified are of ancient origin.** From this, the authors point out that there is likely a large reservoir of unknown tolerance/resistance mechanisms in the environment, and stress the risk if such genes would be transferred to human pathogens, particularly under the exposure to antibiotic residues. **This is perhaps not a really surprising or novel conclusion, but deserves to be stressed and exemplified and backed up more in the literature, as is done here in a nice way.

Many thanks!

Risks for such transfer events (based on genetic context and other factors) are not really discussed in the manuscript, although functionality in E.coli, as demonstrated for several genes, adds to the perceived risk. This observation could be mentioned and discussed together with a range of other factors that influence the risk for gene transfer and eventually treatment failure.

We appreciate the reviewer's comments regarding the level of risk associated with the mobilization of resistance genes from *Paenibacillus* sp. LC231 and we have addressed this on lines 132-133 and 238-242. (All resistance determinants were found on the *Paenibacillus* sp. LC231 chromosome and there is no evidence of nearby mobile elements.) and (No resistance determinant from *Paenibacillus* sp. LC231 is found on mobile elements and therefore the risk of mobilization is low. However, these genes are functional in a heterologous host (*E. coli*) and could be selected for if combined with mobile elements. Our findings are consistent with metagenomics studies of waste water treatment plants²², where most resistance genes are found native to the chromosome.)

The authors speculate on the presence of antibiotic producers in this type of environment, based on a high multi-resistance profile. However, when they compare the resistance profile of the investigated species to the profile of one single related species isolated from another type of environment, the two strains were in fact relatively similar (see supplementary tables 1 and 2). This really does not stress the cave environment as an environment where bacteria are particularly "resistant" (also see below). Hence, the conclusion that the cave environment would be particularly interesting for exploring antibiotic producers is not really substantiated to any large extent by the findings presented in this manuscript. A comparison of resistance or tolerance patterns in a range of bacterial communities from different types of environments, assessed with comparable methodology, would be needed to support such a conclusion. Overall, the "cave" component of the study basically only serves the purpose of providing evidence that the observed "resistance" is intrinsic and not the result from recent changes or human activities.

We welcome the reviewer's criticisms and have expanded/clarified how we reached our hypothesis regarding possible novel antibiotic producers in Lechuguilla Cave on lines 250-258. (While the resistance genes are not unique to *Paenibacillus* sp. LC231, the entire collection is not absolutely conserved among all closely related surface *Paenibacillus*. Therefore, some strains have lost resistance genes in environments that do not select for resistance while LC231 has maintained a full complement of resistance elements. This suggests that the Lechuguilla Cave environment has selected for resistance, and therefore implies the presence of antibiotic producers. It has long been known that soil organisms are excellent sources of antibiotics and we can now hypothesize environments such as Lechuguilla Cave as sites worthy of exploring for antibiotic leads.)

*The authors are unclear about the resistance concept, i.e. they do not define what they mean with "resistance". In the broader literature, resistance is usually referred in either one of two ways: The most common one is defined from a clinical standpoint, that is a level of tolerance to an antibiotic that makes it difficult or impossible to treat an infection with the given antibiotic. This is the basis for the cut-off values for resistance for specific pathogens. What level is defined as resistant from this clinical definition varies between species. For the genus investigated here (*Paenibacillus*), there is no clinical cut-off values to my best knowledge. The other definition commonly used is an ecological one, based on an increased tolerance of a bacterial strain in comparison with another strain of the same species, regardless of the level of increase or the absolute level of antibiotic required to completely stop growth. This is a definition applicable to any cultivable species, but it focus on "acquired" resistance, which is not really the case here. It therefore becomes a bit difficult to use the term resistance in this context where "intrinsic resistance" is studied in a non-pathogenic species. In the author's previous study on bacteria from the same cave, they used a "home-made" definition, with a cutoff of 20 mg/L for all antibiotics (i.e. not*

really comparable to other studies). For some antibiotics, that was considerably below the clinical breakpoints for most pathogens, whereas it was higher for others. **In the present paper, the authors report the actual MIC values for each antibiotic, which is considerably more informative.**

We thank the review for highlighting the difficulties in studying resistance in environmental bacteria. We strongly feel that the best approach to characterizing resistance in *Paenibacillus* sp. LC231 is to compare MIC values to bacteria that are generally considered sensitive (e.g. we used *Staphylococcus aureus* and *Micrococcus luteus* (*Kocuria rhizophila*) in our study). Furthermore, we have clarified our definition of resistance on lines 93-94. (*Paenibacillus* does not have specified MIC values to define resistance. Therefore in this study, resistance is relative to *K. rhizophila* and *S. aureus*.)

On line 95 they state that "26 had an MIC at least 8x higher than susceptible strains", but it is unclear what strains they are talking about. Maybe they mean some other clinically important species?

We have clarified this on lines 94-96. (Both species of *Paenibacillus* had MICs at least 8x higher than *K. rhizophila* or *S. aureus* for 26 antibiotics and, in general, were significantly more resistant to these antibiotics.)

In the reviewer's opinion, the terminology and use of the resistance concept should be clarified and more stringent.

Some genes discovered by functional genomics were not discovered by similarity searches to the CARD database. It should be stressed how dissimilar from the closest resistance genes present in CARD those genes were, and the similarity search strategy applied should be explained.

We appreciate the reviewer's comment and we have expanded on the search strategy used to predict resistance genes. Furthermore, we have included these results in Supplementary Table 3.

Reviewer #2 (Remarks to the Author):

This manuscript (MS) is a very thorough analysis of the resistome of one bacterial strain isolated from a cave which had not been exposed for over 4 million years. There are always concerns over contamination in such cases but I am convinced that sufficient controls were done and are reported in the MS. What is exceptional about this one strain of the genus Paenibacillus is the 18 different resistance mechanisms which were defined in terms of specific chromosomal genes all cloned and expressed in E. coli. No mention is made of mobility so presumably these genes are not readily mobilised?

We thank the reviewer for this comment and have addressed it on lines 132-133. (All resistance determinants were found on the *Paenibacillus* sp. LC231 chromosome and there is no evidence of nearby mobile elements.)

The biochemical analysis of each resistance mechanism has been exemplary and the detail is extremely comprehensive. The authors acknowledge that resistance is an ancient phenomenon which has been discovered in many bacteria from the pre-antibiotic era so these observations in an environmental bacterium are not new. What is new is the description and detail of mechanisms in such an isolate irrefutably isolated for over 4 million years.

Although it is now becoming clear from other published work that there is a wealth of resistance in environmental bacteria distinct from those mechanisms well defined in clinically important bacterial. An excellent example of this is the report in Nat. Comms by Munck et al., 2016 who cloned resistance genes from metagenomic DNA isolated from the bacterial biomass of WWTP digestors. Less than 10% of resistance genes were shared in other metagenomes and many were highly diverse thus supporting the huge reservoir of resistance. They (Munck et al.) did not observe any significant evidence of gene mobilisation except for spectinomycin gene which was mobilised by class 1 integron. I think this may correlate with use of spectinomycin in agriculture. What would be useful to know is how the genes can be mobilised and under what kind of selection if any. Clearly the continued extensive use of antibiotics in agriculture and further dissemination into the environment may act as a catalyst for gene mobilisation. This point is well made and some more comment about mobilisation would be helpful.

We have read the study referenced by the reviewer with great interest, and have incorporated it into our manuscript on lines 238-242. (No resistance determinant from *Paenibacillus* sp. LC231 is found on mobile elements and therefore the risk of mobilization is low. However, these genes are functional in a heterologous host (*E. coli*) and would be selected for if combined with mobile elements. Our findings are consistent with metagenomics studies of waste water treatment plants²², where most resistance genes are found native to the chromosome.)

Reviewer #3 (Remarks to the Author):

*In this paper, Pawlowski et al perform an extensive analysis of the antibiotic resistance phenotype of *Paenibacillus* sp. LC231, a strain that was isolated from Lechuguilla Cave, which forms an underground ecosystem that has been isolated for millions of years. **Through an elegant combination of microbiology, microbial genomics and chemistry**, the authors show that the *Paenibacillus* strains carries 18 antibiotic resistance determinants (including 5 new enzyme families and 3 novel resistance determinants). **These data show that bacteria from pristine environments can be rich in resistance genes, providing evidence (beyond other publications of the Wright group and others) that the presence of antibiotic resistance genes are natural phenomena.** There is no evidence that the 'ancient' cave strain of *Paenibacillus* is, in any way, remarkable as *Paenibacillus* strains from other (non-isolated) sites can have very similar antibiotic resistance profiles.*

While I have no reason to doubt the validity of the approach or the quality of the data, I believe some aspects (mainly in terms of the presentation of the data) will need to be improved or corrected. These are specifically outlined below.

There are no issues with the statistical analysis of the data in this manuscript and references are appropriate. The manuscript is generally well-written but contains a few errors in language or style (outlined below).

The conclusions (particularly for the resistance determinants that were biochemically characterized) of resistance determinants of Paenibacillus LC231 are supported by the data.

However, I do not agree with the statement that the data indicate that 'new antibiotic producers may be present in Lechuguilla Cave' (l. 35). I fail to see why Lechuguilla Cave should contain new antibiotic producers and why it is singled out as a site 'worthy of exploration for antibiotic leads' (l. 244). While it is certainly possible that new antibiotic producers can be found in the cave, there is no reason to believe that the cave is a more promising environment than other sites (soil, water etc) to find antibiotic producers. I feel these statements should be removed and the conclusions should focus on the in-depth characterization of the resistance gene repertoire of Paenibacillus LC231.

We thank the reviewer for their constructive criticism regarding antibiotic producers in Lechuguilla Cave, and have used it as an opportunity to expand and clarify our conclusion. We have carefully reworded our statements on lines 250-258 with evidence to support our conclusions as discussed above in the context of Reviewer 1's comments. (While the resistance genes are not unique to *Paenibacillus* sp. LC231, the entire collection is not absolutely conserved among all closely related surface *Paenibacillus*. Therefore, some strains have lost resistance genes in environments that do not select for resistance while LC231 has maintained a full complement of resistance elements. This suggests that the Lechuguilla Cave environment has selected for resistance, and therefore implies the presence of antibiotic producers. It has long been known that soil organisms are excellent sources of antibiotics and we can now hypothesize environments such as Lechuguilla Cave as sites worthy of exploring for antibiotic leads.)

Suggested improvements:

l. 34. 'Longevity of these genes in this environment'. I believe the authors refer to the cave system with 'this environment' but the data point towards a natural reservoir for resistance genes in Paenibacillus inside or outside the cave. Therefore 'this environment' should be replaced by 'this genus' or this phrase should be removed.

We have made this change as suggested.

l. 61. Combinatorialization: why not simply 'combination of multiple genes on a single genetic platform' or similar?

We have made this change on line 59-61. (Furthermore, the diversity of resistance in individual strains and the combination of multiple genes on a single genetic platform are reaching new heights.)

l. 94 - 98. This section is somewhat confusing. Subjective terms like 'highly resistant' should be avoided and data should be described in a more objective, quantitative fashion. Antibiotics do

not have an MIC ('26 had an MIC at least 8x higher', which is a difficult-to-understand phrase anyway) but strains have an MIC for specific antibiotics.

We appreciate the reviewer's comments and have carefully reworded this statement on lines 94-96. (Both species of *Paenibacillus* had MICs at least 8x higher than *K. rhizophila* or *S. aureus* for 26 antibiotics and, in general, were significantly more resistant to these antibiotics.)

*I believe the comparison with 'susceptible strains' is done with *M. luteus* and *S. aureus* but these data are not included in the manuscript, so it is not possible to check for which antibiotics the MICs are {greater than or equal to}8-fold higher. I suggest Supplementary tables 1 and 2 are to be combined in a single table and the data for *S. aureus* and *M. luteus* are added.*

We recognize that the large amount of data regarding the resistance phenotype of *M. luteus* (*K.rhizophila*), *S. aureus* and *P. lautus* ATCC 43898 is better presented by incorporating Supplementary Figure 1 (MIC data for *K.rhizophila* and *S. aureus*) into Supplementary Table 2 (MIC data for *P. lautus* ATCC 43898). It is our opinion that combining Supplementary Tables 1 and 2 would result in a large and confusing table and is better presented separately.

The statement on inactivation of antibiotics is, at this point in the manuscript, not supported by the data.

Now explicit in Supplementary Table 1.

l. 99 - 115. In this section (putative) resistance genes are detected using genome sequencing followed by interrogation of the CARD database. The correlations between resistant phenotypes and detection of resistance determinants are not always convincing. The authors mention 'decreased rifampin sensitivity' but the MIC for this antibiotic is only 1 µg/ml, which is generally considered sensitive.

While we appreciate the reviewer's comments, we believe that environmental resistance is relative and we consider a difference in MIC of 64x to *M. luteus* and *S. aureus* to be significant despite an MIC of 1 µg/mL.

The role of rph has not been substantiated so l. 104 should read 'may inactivate'. 'These drugs' (l. 105) should be corrected to 'this drug'.

We thank the reviewer for identifying this and we have make this change as suggested.

*The role of the Cfr-like gene is also not clear. How can the authors be so sure that it is responsible for the slightly decreased sensitivity to a wide range of antibiotics from different classes. The gene appears not have been heterologously expressed in *E. coli* (Table 1). I believe these experiments should be performed to provide additional evidence for the role of the cfr gene in LC231.*

We have modified our statement to clarify that the Cfr-like gene has already been studied by another group on line 105-109. (Our informatic analysis identified an orthologue of the 23S rRNA methyltransferase Cfr (95% identical to the characterized Cfr-like enzyme CIPa from *Paenibacillus* sp. Y412MC40¹⁷), which is consistent with slightly decreased sensitivity to

linezolid, pleuromutilin, streptogramin A, lincosamide and chloramphenicol antibiotics.) We have also modified Supplementary Table 1 to indicate that the gene is Cfr-like and not a Cfr.

As with rifampin, the resistance phenotypes are not at all clear (e.g. MIC for linezolid of LC231 is 1 µg/ml, which would generally be considered susceptible; clindamycin resistance is linked to cfr in supplementary table 1, but is not discussed in text).

Finally, as the authors have a complete genome sequence, I feel they should add information on the location (chromosome, plasmid) and/or genetic context of the resistance gene as it is interesting to know whether resistance genes appear to be fixed on the chromosome or are carried on mobile genetic elements.

We recognize that mobility of resistance genes in *Paenibacillus* sp. LC231 is important to address and we have done so now on lines 132-133 and 238-242. (All resistance determinants were found on the *Paenibacillus* sp. LC231 chromosome and there is no evidence of nearby mobile elements.) and (No resistance determinant from *Paenibacillus* sp. LC231 is found on mobile elements and therefore the risk of mobilization is low. However, these genes are functional in a heterologous host (*E. coli*) and could be selected for if combined with mobile elements. Our findings are consistent with metagenomics studies of waste water treatment plants²², where most resistance genes are found native to the chromosome.)

Genome sequence data should be submitted to NCBI or ENA and an accession number should be provided in the manuscript.

We have submitted the *Paenibacillus* sp. LC231 genome sequence to GenBank, edited the GenBank WGS annotation to include each resistance gene as well as submit each resistance gene sequence individually to ensure that the community can easily access this information. Please see lines 284-288 (*Nucleotide sequences for the Paenibacillus sp. LC231 genome have been deposited in GenBank with the accession code JFOM000000000*). Each resistance determinant studied was deposited with the following accession codes KX531043, KX531044, KX531045, KX531046, KX531047, KX531048, KX531049, KX531050, KX531051, KX531052, KX531053, KX531054, KX531055, and KX531056. Additionally, each resistance determinant was deposited in CARD.).

l. 131 - 132: I note that not all 'new resistance elements' are discussed in the manuscript (e.g. TetAB(48) and TaeE) so this line should be rephrased.

We have made the suggested modification.

l. 154, l. 190. Add 'e-value: ' to the score for clarity

We agree and have made the suggested modification.

l. 177 - 178. Please add quantitative information on the relatedness of MphI to other macrolide kinases.

Supplementary Table 3 includes quantitative information on the relatedness of all resistance enzymes in this study.

l. 193 - 194: 'confer' should be corrected to 'confers'.

We have made the suggested change.

What is the mechanism for the increased sensitivity to linezolid and pleuromutilins? E. coli is intrinsically resistant to these antibiotics.

While we agree that identifying the mechanism of increased sensitivity to linezolid and pleuromutilins is interesting, we believe that it is beyond the scope of this paper.

l. 201. Suggest to replace 'track with' with 'match'

We have made this modification.

l. 204. It is not entirely clear to what gene/organism the Vat protein from P. lactis is 94% identical.

We have clarified this on line 205. (We found that *Paenibacillus vortex* and *Paenibacillus* sp. FSL R5-808 are the most closely related strains to *Paenibacillus* sp. LC231 but their Vat enzymes are only 55% identical to VatI. In contrast, the Vat from *Paenibacillus lactis* is 94% identical to VatI.)

l. 208. P. lautus is not present in Fig. 5, presumably because no genome sequence is publicly available. However, it is of considerable interest to determine the presence of resistance genes of this organism as well, as it appears closely related to LC231 (in addition, please write 16S rRNA gene sequence in l. 208). As this strain is modern (isolated from the intestinal tract of a child), it shows that LC231 is not particularly different from currently circulating Paenibacillus strains.

Paenibacillus lautus ATCC 43898 was isolated from the intestinal tract of a child in 1919 and sequencing this genome would permit the identification of the resistance genes in this strain. Another group has already submitted an NCBI BioProject study regarding sequencing this genome (accession PRJDB1365), but the sequence is not yet released.

l. 211. It is perhaps better to write 'a clade within the genus Paenibacillus' to highlight that not all paenibacilli share the multi-drug resistant phenotype (see Fig 5A).

We agree that we could make this clearer in the text, and not only make the distinction in Figure 5. Please see lines 210-213. (We generated a quantitative antibiogram for this organism that was very similar to that of *Paenibacillus* sp. LC231 from Lechuguilla Cave, demonstrating that this multi-drug resistant phenotype is native to a clade within the *Paenibacillus* genus Supplementary Table 2).)

l. 242. What is meant with 'resource competitive'

We have removed this statement.

l. 248. Delete 'remarkable and'

We have made the suggested edit.

l. 249. I am not sure if one can claim that the environmental reservoir of resistance is 'underappreciated'.

We have removed this statement.

*l. 249 - 252. Perhaps the authors can shortly discuss the likelihood of the resistance genes from LC231 being mobilized to pathogens (e.g. are they located on plasmids? See comments above). In addition, it could be said that these resistance genes are unlikely to be particularly dangerous, even if they end up in opportunistic pathogens like *Staphylococcus aureus* and *Klebsiella pneumoniae* as most appear to have a remarkably narrow substrate range and most do not appear to target clinically important antibiotics: this could also be discussed here.*

We appreciate the review's comments regarding the risk of mobilization and have incorporated it into the manuscript on lines 132-133 and 238-242. (All resistance determinants were found on the *Paenibacillus* sp. LC231 chromosome and there is no evidence of nearby mobile elements.) and (No resistance determinant from *Paenibacillus* sp. LC231 is found on mobile elements and therefore the risk of mobilization is low. However, these genes are functional in a heterologous host (*E. coli*) and could be selected for if combined with mobile elements. Our findings are consistent with metagenomics studies of waste water treatment plants²², where most resistance genes are found native to the chromosome.)

l. 288: Cedarlane: add city, country.

We have now included this information.

*l. 297: What is the strain name of the *M. luteus* isolate used in this study? If it is ATCC 9341, it should be renamed as *Kocuria rhizophila* (see Tang & Gilbert. *Int J Syst Evol Microbiol* 2003 53:995)*

The *M. luteus* strain we use is a lab strain we have used for two decades. We have sequenced the 16S rRNA and indicated percent identity to *Kocuria rhizophila* ATCC 9341 on lines 313-315. (*Staphylococcus aureus* RN4220 was cultured on LB agar, and *Staphylococcus saprophyticus* ATCC 15305 and *K. rhizophila* (16s rRNA is 99% identical to *K. rhizophila* ATCC 9341) were cultured on TSA.)

l. 300 - 301: the reference should be put after 'broth microdilution method' and correct 'modification' to 'modifications'.

We have made the suggested modifications.

l. 306. The abbreviation MHB (for Muller Hinton Broth) is not explained

We have now defined the abbreviation on line 324.

l. 347. Write 'strains' or 'isolates' after Paenibacillus.

We have added the suggested modification.

Fig 2: The active version of bacitracin is the X = -NH₂ version. Please correct.

We apologize for this error and very much appreciate the reviewer noticing this. We have regenerated Figure 2 with the correct structure of bacitracin.

Fig 3: The panels of this figure have been switched. Terms like multi-bond HMBC, TOCSY and HSQC correlations are not explained in the manuscript and are difficult to follow to readers without expertise in NMR.

We appreciate the reviewer identifying this error and we have modified the figure legend to correctly reflect the figure panels.

Fig 5. I assume the dendrogram-type tree a representation of the phylogenetic tree in Fig 6? If yes, please specify this in the legend. Please write 'An assembly gap in RphB is indicated with an asterisk'.

We have included the review's suggestions into the figure legend for Figure 5.

Table 1: the MICs of the wild-type strain and the strains expressing the resistance determinants should be shown (rather than a fold change in MIC). I note that a fold change of 0 (for mphI and erythromycin) is most likely a mistake (no change in resistance is a fold change of 1).

We have updated Tables 1 and Supplementary Table 10 to include this information

Supplementary information:

l. 29 - 30. I believe this figure can be deleted if my comments on Table S1 and S2 are addressed. It is also not clear what the authors mean with 'a value equal-to or greater-than is indicated': equal-to, greater-than what?

We have included Supplementary Figure 1 into Supplementary Table 2. Please see our response to this reviewers comment above.

What are 1x Cpa buffer and 1x Mph buffer: presumably the buffer described in l. 267 and l. 268? Please specify. For the experiments in l. 326 - 373 it is also not clear in which buffers these experiments were performed (presumably the same as outlined earlier, e.g. l. 267 - 271?): this should be specified.

We apologize that this wasn't made clear, and in response we have included buffer information at each point in the materials and methods section (lines 266-267, 294-295, 320-321, 326-327, 333, 339-340, 346-347, 352, 357-358 and 366-367.)

In l. 326 - 373, the phrase 'Km was determined as follows' followed by the concentration of reagents is insufficient to reproduce the experiment and should be rewritten.

We have included more information on lines 314-369

REVIEWERS' COMMENTS:

Reviewer #1 (Remarks to the Author):

Review of revised version of Pawlowski et al by reviewer #1

With regards to my point on a discussion of factors influencing risks for gene transfer to pathogens, the authors refer to lines 132-133 and 238-242. The authors only bring up chromosomal location in the absence of adjacent known mobile elements (points to less risk), and functionality in a model pathogen (points to higher risk). They still do not mention aspects such as whether these bacteria have ample opportunities to interact with human pathogens (ecological connectivity) which in practice is almost absent in the cave!, costs of the resistance genes (not known, I guess), known selection pressure favoring bacteria taking up the genes (I do not think there is any proof for that, see also below) etc. The discussion of risk is shallow as it stands now.

I also raised a point in that the findings in this study do not provide any strong support for a selection pressure from antibiotics producers in the cave, and that therefore this does not motivate the suggestion that this cave would be particularly promising for looking for novel antibiotics producers in comparison with other environments. Please note that exactly the same critique was brought up independently by reviewer #3. The authors argue for their position and now bring up the finding that that other members of this species do not carry all of the resistance genes found in the cave strain and that they have been lost in other environments not selecting for resistance. First of all, if we look at what the authors refer to as group 1 in figure 5, there are other strains that indeed do contain all of these genes and some that contain almost all of these genes. Secondly, the comparison is based from the standpoint of comparing if the specific genes present in the cave-strain are also found in the other strains/species. Not the reverse. I.e. the authors do not investigate if there are other "resistance genes" present in the other strains that are not present in the cave strain. With this type of comparison, the strain that you base your comparison on (here the cave strain) will always be the one carrying the highest number of genes (all), hence the observation that some other strains lack some of the genes is not evidence or even support for what the authors claim. Third, they talk about "loss of genes" in some strains. I do not see the support to why the ancestors should have possessed the gene and there is a loss in the other strains, it could have been a gain in the cave strain? Fourth, the authors talk about loss of resistance genes in "environments that do not select for resistance". I argue that the authors show no evidence that there is a specific selection pressure for these forms of resistance in the cave either, at least not to any higher degree than in other environments, hence the explanation lacks support. Taken together, I still think the authors should drop the speculations that the data in this study supports the idea that the cave environment is a particularly interesting to look for antibiotic producers, at least the argument raised are not sufficient.

The authors have clarified that they based their definition of resistance on comparison with the susceptibility of two other species. They "strongly feel" this is the best approach. I disagree. Then the definition of resistant is strongly dependent on which two species (or even strains) you happen to compare with. I indicated in my previous response how resistance usually is defined from a clinical or ecological standpoint. The authors have not clarified or motivated their use of the resistance concept in an acceptable way.

Overall, I think the study is good, but the responses to the (few) points I brought up was not satisfactory, hence I think the manuscript still needs revision.

Reviewer #3 (Remarks to the Author):

I feel the authors have convincingly addressed the comments of the reviewers.

I would prefer the authors to write '18 chromosomally located resistance elements' in l. 31 to highlight the low potential for mobilization of these resistance genes.

REVIEWERS' COMMENTS:

Reviewer #1 (Remarks to the Author):

Review of revised version of Pawlowski et al by reviewer #1

With regards to my point on a discussion of factors influencing risks for gene transfer to pathogens, the authors refer to lines 132-133 and 238-242. The authors only bring up chromosomal location in the absence of adjacent known mobile elements (points to less risk), and functionality in a model pathogen (points to higher risk). They still do not mention aspects such as whether these bacteria have ample opportunities to interact with human pathogens (ecological connectivity) which in practice is almost absent in the cave!, costs of the resistance genes (not known, I guess), known selection pressure favoring bacteria taking up the genes (I do not think there is any proof for that, see also below) etc. The discussion of risk is shallow as it stands now.

We have expanded our discussion of risk on lines 239-243. Furthermore, we did not study any fitness costs associated with expressing these resistance genes and therefore did not comment on it.

I also raised a point in that the findings in this study do not provide any strong support for a selection pressure from antibiotics producers in the cave, and that therefore this does not motivate the suggestion that this cave would be particularly promising for looking for novel antibiotics producers in comparison with other environments. Please note that exactly the same critique was brought up independently by reviewer #3. The authors argue for their position and now bring up the finding that that other members of this species do not carry all of the resistance genes found in the cave strain and that they have been lost in other environments not selecting for resistance. First of all, if we look at what the authors refer to as group 1 in figure 5, there are other strains that indeed do contain all of these genes and some that contain almost all of these genes. Secondly, the comparison is based from the standpoint of comparing if the specific genes present in the cave-strain are also found in the other strains/species. Not the reverse. I.e. the authors do not investigate if there are other "resistance genes" present in the other strains that are not present in the cave strain. With this type of comparison, the strain that you base your comparison on (here the cave strain) will always be the one carrying the highest number of genes (all), hence the observation that some other strains lack some of the genes is not evidence or even support for what the authors claim. Third, they talk about "loss of genes" in some strains. I do not see the support to why the ancestors should have possessed the gene and there is a loss in the other strains, it could have been a gain in the cave strain? Fourth, the authors talk about loss of resistance genes in "environments that do not select for resistance". I argue that the authors show no evidence that there is a specific selection pressure for these forms of resistance in the cave either, at least not to any higher degree than in other environments, hence the explanation lacks support. Taken together, I still think the authors should drop the speculations that the data in this study supports the idea that the cave environment is a particularly interesting to look for antibiotic producers, at least the argument raised are not sufficient.

We have removed these statements.

The authors have clarified that they based their definition of resistance on comparison with the susceptibility of two other species. They "strongly feel" this is the best approach. I disagree. Then the definition of resistant is strongly dependent on which two species (or even strains) you happen to compare with. I indicated in my previous response how resistance usually is defined from a clinical or ecological standpoint. The authors have not clarified or motivated their use of the resistance concept in an acceptable way.

Paenibacillus lacks established genetic tools and we could not use gene-deletion studies to investigate the phenotypes within this strain. Therefore, we could not use either of the definitions of resistance that this reviewer suggested. We have clarified the object of our study on lines 94-96.

Overall, I think the study is good, but the responses to the (few) points I brought up was not satisfactory, hence I think the manuscript still needs revision.

Reviewer #3 (Remarks to the Author):

I feel the authors have convincingly addressed the comments of the reviewers.

I would prefer the authors to write '18 chromosomally located resistance elements' in l. 31 to highlight the low potential for mobilization of these resistance genes.

We have added this on line 33.